# Representation Learning via Consistent Assignment of Views over Random Partitions

**Thalles Silva**
Institute of Computing
University of Campinas
`thalles.silva@students.ic.unicamp.br`

**Adín Ramírez Rivera**
Department of Informatics
University of Oslo
`adinr@uio.no`

## Abstract

We present Consistent Assignment of Views over Random Partitions (CARP), a self-supervised clustering method for representation learning of visual features. CARP learns prototypes in an end-to-end online fashion using gradient descent without additional non-differentiable modules to solve the cluster assignment problem. CARP optimizes a new pretext task based on random partitions of prototypes that regularizes the model and enforces consistency between views' assignments. Additionally, our method improves training stability and prevents collapsed solutions in joint-embedding training. Through an extensive evaluation, we demonstrate that CARP's representations are suitable for learning downstream tasks. We evaluate CARP's representations capabilities in 17 datasets across many standard protocols, including linear evaluation, few-shot classification, $k$-NN, $k$-means, image retrieval, and copy detection. We compare CARP performance to 11 existing self-supervised methods. We extensively ablate our method and demonstrate that our proposed random partition pretext task improves the quality of the learned representations by devising multiple random classification tasks. In transfer learning tasks, CARP achieves the best performance on average against many SSL methods trained for a longer time.

## 1 Introduction

Learning from unlabeled data has been one of the main challenges in computer vision. Recent approaches based on self-supervised learning (SSL) have significantly reduced the gap between supervised and unsupervised pre-trained representations. Nowadays, self-supervised pre-training on vast quantities of unlabeled data, prior to learning a downstream supervised task of interest, can be more effective than supervised pre-training for many tasks [7, 19, 22].

Current SSL methods can be divided into two classes: (1) self-supervised embedding prediction [9, 24, 39, 41] and (2) clustering [1, 5, 6, 29]. As the name suggests, embedding prediction methods work directly in the representation space. They are trained with either contrastive or non-contrastive loss functions. Instead of reconstructing the input signal, their loss function maximizes agreement between embeddings of the same view and optionally pushes representations from different views apart. On the other hand, clustering methods discretize the representation space by learning a finite set of prototypes. These prototypes aggregate representations from different images that are similar enough to be assigned together. Nevertheless, recent SSL methods build a joint-embedding architecture that can be pure siamese [4] or use different encoders.

---

Code at `https://sthalles.github.io/carp/`.

37th Conference on Neural Information Processing Systems (NeurIPS 2023).

The most challenging and significant difference among these methods is how they avoid trivial solutions when training joint-embedding architectures. Contrastive methods avoid trivial solutions by explicitly pushing representations of negative samples away from the anchor representation, while non-contrastive methods avoid trivial solutions by regularization or architectural designs [10, 22].

Among self-supervised clustering methods, recent work proposed avoiding trivial solutions by using non-differentiable modules such as the Sinkhorn-Knopp algorithm [1, 7, 14] and classic machine learning methods [29, 42] such as $k$-Means clustering or $k$-Nearest Neighbor, to solve the cluster assignment problem.

Consistent assignments [37, 38] were recently proposed as a way to learn prototypes to improve the representations. Similarly, our learning objective imposes two constraints (1) consistent assignment of views over learnable prototypes and (2) a uniform distribution for the average predictions within a batch. However, we show that such a strategy does not scale well to enormous datasets containing millions of classes. In such situations, we need to model a large number of prototypes while enforcing consistent assignment between views and avoiding collapsed solutions. We show that a naive implementation of this strategy makes the learning problem challenging from a training stability perspective, where the model quickly settles for a trivial solution by assigning all views' embeddings to the same prototype.

To overcome these issues, we propose a novel self-supervised approach based on the consistent assignment of views over random partition sets (CARP). We train CARP to minimize a consistency loss, which encourages the model to assign different views of the same unlabeled example to the same prototype. We solve the dimensionality problem by enforcing smaller pseudo-classification problems through the introduction of random partitions that enforce consistency and regularize the model. The energy between the views' representations and the trainable prototypes (within random partitions) allows us to automatically bootstrap predictions and targets to our consistency loss. Our contributions are three-fold:

1. A novel and entirely online joint-embedding learning strategy based on self-supervised clustering, see Figure 1. We propose a divide-and-conquer pretext task based on randomly generated partitions of learnable prototypes. Our loss function allows stable training of joint-embedding architectures in a self-supervised context.
2. A framework that simplifies self-supervised training and does not require normalization techniques [7, 9] or the necessity of mining negatives for contrastive training [9, 31, 32].
3. A differentiable assigner module that generates soft pseudo-labels by comparing the representations of image views to prototypes within random partition subsets. To avoid trivial solutions, we enforce the average predictions over a batch to be non-informative over the set of prototypes within a random subset.

## 2   Related work

**Self-supervised embedding prediction methods** operate directly in the representation space by learning a metric such that embeddings from views of the same image are closer to one another while embeddings from views of different images are far away in the feature space. These methods can be trained using contrastive or non-contrastive loss functions. Methods that minimize a loss function with a contrastive term date back to 1990s [4, 13, 21]. They must explicitly find representations from non-correlated images to use as negatives. Recent contrastive methods include InstDisc [46], CPC [32], SimCLR [9] and MoCo [11, 24]. These methods learn unsupervised representations by minimizing nearly the same contrastive loss function, i.e., the InfoNCE [32]. CARP does not directly optimize the views' embeddings, nor is it a contrastive method. Instead, we learn a set of general prototypes using a random partition strategy that stabilizes the learning process and avoids trivial solutions commonly found when training joint-embedding SSL models.

On the other hand, non-contrastive methods work by approximating embeddings of views from the same image. The main advantage is not requiring explicit opposing representations in the loss formulation. To avoid trivial solutions, a common approach is to implement a "stop-gradient" operation that prevents the gradient signal from flowing to the two branches of the join-embedding architecture simultaneously. BYOL [22] and SimSiam [10] are examples of such methods. CARP takes advantage of non-contrastive training since it does not require mining negatives for optimization. Also, different from Grill et al.'s [22] work, CARP does not require a momentum encoder, though using it

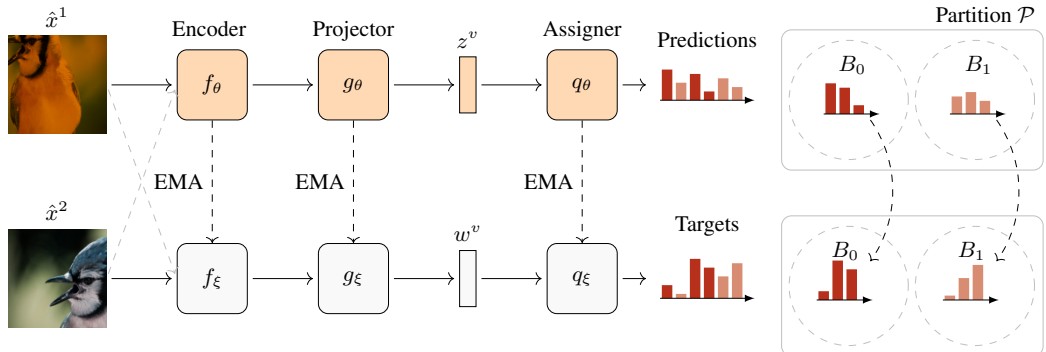

Figure 1: CARP's training architecture. From two views, we train an encoder $f_{(\cdot)}$, followed by a projection $g_{(\cdot)}$ to produce representation vectors $z^v$ and $w^v$, respective to the parameters $\theta$ and $\xi$ for each branch, for each view indexed by $v$. The representations are fed to an assigner function $q_{(\cdot)}$ that produces normalized distributions of views w.r.t. the learnable prototypes. Note that $\theta$ are the trainable weights, and $\xi$ are an exponential moving average of $\theta$. We create partitions by randomly arranging the prototypes into a predefined number of blocks. E.g., from a total of $K = 6$ prototypes, we create $N_{\mathcal{P}} = 2$ blocks, each containing $N_B = 3$ prototypes. Then, we enforce consistent assignment of views over prototypes within the blocks.

significantly improves the learned representations. CARP trains a joint-embedding architecture and uses the "stop-gradient" operation in conjunction with a regularized pretext task based on random partitions of prototypes to avoid mode collapse.

**Self-supervised clustering methods** do not work directly on the views' embeddings. Instead, they learn a set of prototypes that are used to solve subsequent pretext tasks in different ways. Caron et al. [5], for instance, used $k$-Means clustering at every epoch to cluster the representations from the entire dataset and produce pseudo-labels for the training images. Then, a classifier head is trained to predict the pseudo-labels. Caron et al. [6] proposed a method to combine the rotation prediction pretext task [18] with clustering. Li et al. [29] presented a method based on expectation-maximization (EM) that merges clustering with the contrastive learning framework from He et al. [24]. Recent work by Caron et al. [7] and Asano et al. [1] combine SSL with clustering. They utilize the non-differentiable Sinkhorn-Knopp algorithm to solve the cluster assignment problem without falling into collapsed solutions. Silva and Ramírez Rivera [38] proposed, CARL, an online clustering method that does not require non-differential algorithms to avoid trivial solutions.

**Contrast to previous approaches.** Instead of solving the cluster assignment problem using a non-differentiable algorithm such as the Sinkhorn-Knopp, CARP is trained end-to-end with gradient descent. Different from Caron et al.'s [8] work, our model does not require extra momentum encoders or data structures to store previous predictions as a way to avoid trivial solutions. Unlike CARL [37, 38], CARP avoids trivial solutions by posing the optimization problem at the level of random partitions of prototypes that solve the high-dimensionality nature of the task. Unlike Caron et al.'s [6] work, our method does not require clustering the entire dataset every epoch to generate pseudo-labels. Instead, CARP generates soft pseudo-labels in an online fashion from examples in a single minibatch.

Currently, the best self-supervised methods [8, 12] use vision transformers [15, 44] as their backbones. Recent methods [20] employ a masking pretext task where some patches of the views are manually hidden, and the network is tasked to infer the missing pieces. In this paper, we consider transformer-based methods to be out-of-scope of our compared backbones. Thus, we do not include them in our results to maintain a fair comparison.

## 3 Consistent Assignment of Views

From an image $x_i$, we create two views, $\hat{x}_i^1 = T(x_i)$ and $\hat{x}_i^2 = T(x_i)$, using a stochastic function $T$ that applies a set of random image transformations to $x_i$ (cf. Appendix C.1). CARP is a joint-embedding architecture with two modules: a differentiable (student) and a non-differentiable (teacher)

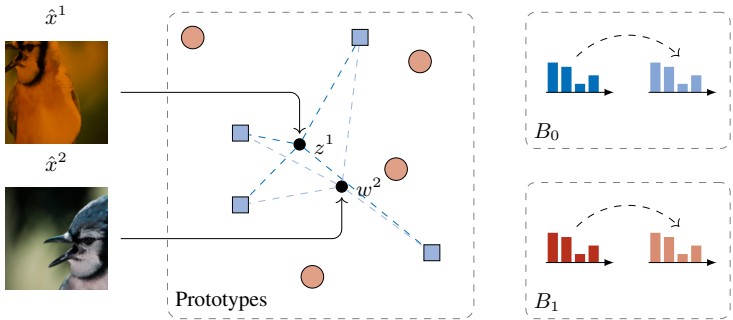

Figure 2: Instead of posing a pseudo-classification problem overall prototypes (circles and squares), the views are assigned (colored dashed lines) to a subset of prototypes (blocks), devising multiple pseudo-classification problems. Then, we contrast their distributions of assignments. (For this example, $K = 8$, $N_B = 4$ with $N_{\mathcal{P}} = 2$.)

branch. Each module has its own set of weights and the same architectural design. Both contain an encoder $f_{(\cdot)}$ and a projection head $g_{(\cdot)}$. The differentiable student receives a pair of views and produces embedding vectors $z_i^v = g_\theta(f_\theta(\hat{x}_i^v))$, for $v \in \{1, 2\}$. Similarly, the non-differentiable teacher produces target embeddings $w_i^v = g_\xi(f_\xi(\hat{x}_i^v))$.

The objective is to learn a set of prototype vectors $C$ to discretize the embedding space. These prototypes are not meant to represent the true classes of the data. Instead, they may be interpreted as anchors to attract views of a given image to a commonplace in the embedding space. The function $q(\cdot, \cdot)$ receives the views' representations, $z_i^v$ and $w_i^v \in \mathbb{R}^{1 \times d}$, as input and outputs normalized probability vectors relating the views' embeddings with the prototypes such that $s_i^v = q(z_i^v, C)$ and $t_i^v = q(w_i^v, C)$, where $s_i^v$ and $t_i^v \in \mathbb{R}^{1 \times K}$ are the normalized probabilities of a view, $\hat{x}_i^v$, w.r.t. the prototypes $C \in \mathbb{R}^{K \times d}$. Note that $d$ is the dimensionality of the embedding vector, $K$ is the number of prototypes, and the assigner $q(h, C) = \text{softmax}\left(h \cdot C^T\right)$.

To avoid trivial solutions in the joint-embedding training, we need a loss function that prevents the assignment of all representation vectors $z_i$ to a unique prototype. Unlike previous work, we seek a method that solves the cluster assignment problem in an online fashion using gradient descent.

We propose a loss function composed of two terms: consistency and entropy. The consistency term learns the relations between embedding vectors and prototypes. It enforces different views of the same image to be assigned to the same prototype with high confidence. For normalized probability vectors $a$ and $b$, we define the consistency term as

$$\mathcal{L}_c(a, b) = -\log \langle a, b \rangle, \tag{1}$$

where $\langle \cdot, \cdot \rangle$ is a dot product.

The consistency loss is optimized when the two views $\hat{x}_i^1$ and $\hat{x}_i^2$ are assigned to the same prototype with maximal confidence, i.e., when the probability distributions of the two views $s_i^1$ and $s_i^2$ resemble equal one-hot vectors.

If we optimize the consistency loss, $\mathcal{L}_c$, training collapses to a state where all views are assigned to the same prototype. A common approach to avoid such failure is to ensure that all the prototypes get roughly the same number of assignments. Let us define the function

$$\text{avg}\left(\{(a_i, b_i)\}_{i=1}^L\right) = \frac{1}{L} \sum_{i=1}^L a_i + b_i \tag{2}$$

as the average probability across the representations within a batch of size $L$. For our distributions, we define $\bar{p} = \text{avg}(\{(s_i^v, t_i^v)\}_{i=1}^N)$. If we maximize the entropy of the mean probabilities of a batch, $H(\bar{p})$, we will encourage the average predictions to be closer to a uniform distribution. Previous work [2, 26, 42] has used this entropy term in various scenarios, ranging from discriminative unsupervised clustering to semi-supervised learning. In our case, this term enforces a non-informative prior over the prototypes with a particular schedule learning. Thus, the final proposed objective to

minimize is

$$\mathcal{L} = \frac{1}{N} \sum_i^N \left( \mathcal{L}_c(s_i^1, t_i^2) + \mathcal{L}_c(s_i^2, t_i^1) \right) - \lambda_e H(\bar{p}), \tag{3}$$

where $\lambda_e > 0$ trades off consistency at the view level with the average uniform assignment at the batch level. Note that the contribution of the entropy term decays as training progresses.

### 3.1 Limitations of Consistent Assignments

The formulation of the consistent assignments (3) is similar to CARL [37, 38], except for the additional teacher stream that stabilizes training and improves performance. Nevertheless, training an unsupervised system by minimizing the loss (3) is challenging. The main limitation is how to avoid trivial solutions in unsupervised training of joint-embedding architectures. For some cases, tuning the contribution of the entropy term might be enough to optimize the loss (3) stably. However, adjusting such a hyperparameter is difficult because one configuration does not hold for all training conditions. For instance, if the value of $\lambda_e$ is too small, the consistency term wins the arms race, and the average distribution over the batch, $H(\bar{p})$ becomes one-hot alike, i.e., all views end up assigned to the same prototype. If the value of $\lambda_e$ is too large, the entropy term gets the upper hand, and collapse is avoided. However, the process of view assignment is neglected over the policy of distributing views uniformly, which results in poor performance of the learned representations, as shown by Table B.2.

For a small number of general prototypes, training is more stable, and the model avoids collapse with a simple tuning of the $\lambda_e$ parameter. However, for a larger number of general prototypes, stability becomes an issue. The main problem lies with the entropy term. When the distribution is larger, i.e., $K \gg N$, regular batch sizes, such as $N = 64$ or $N = 128$, become too small to properly model the distribution, i.e., the signal is too weak for most prototypes. Consequently, to avoid collapse, we need to increase the strength of the entropy term or increase the batch size, which in turn decreases performance.

To address such limitation, we propose to decouple the loss function (3) into smaller sub-problems. Instead of enforcing both consistency and uniform assignments over all the general prototypes, we propose a pretext task over subsets or blocks of a random partition of the general prototype set $C$.

## 4 Assignment based on Random Partitions

Given the set of $K$ trainable prototypes $C = \{c_1, c_2, ..., c_K\}$, we define a partition of $C$ as $\mathcal{P} = \{B_i \subset C\}_{i=1}^{N_{\mathcal{P}}}$, such that $\emptyset \notin \mathcal{P}$, $\bigcup_i B_i = \mathcal{P}$ where $B_i \in \mathcal{P}$, and $B_i \cap B_j = \emptyset$ for all $B_i, B_j \in \mathcal{P}$, and $i \neq j$. We refer to $B_i$ as a block or subset of the partition. We are interested in a partition set $\mathcal{P} = \{B_i\}_{i=1}^{N_{\mathcal{P}}}$ of size $N_{\mathcal{P}}$, i.e., $|\mathcal{P}| = N_{\mathcal{P}}$.

Using the concept of a partition of a set, we can define a framework of pretext tasks over partition blocks that satisfies the learning problem defined in Section 3. If the size of a partition block, $B_i$, equals the number of prototypes, $N_B = K$ then the partition $\mathcal{P}$ is trivial, i.e., $\mathcal{P} = \{B_1\} = \{C\}$. If the size of the partition blocks equals $N_B = 1$, then we have $K$ blocks in $\mathcal{P}$, and each block has a unique prototype. Here, the learning task is equivalent to multiple binary classification problems, where each output score, if normalized, expresses the likelihood of a data point $x_i$ to independently belong to each prototype.

However, if the block size $1 < N_B < K$, and $N_B$ divides $K$, then the partition $\mathcal{P}$ will be composed of $N_{\mathcal{P}} = \lfloor K/N_P \rfloor$ blocks. We define $\mathcal{P}$ by randomly assigning $N_B$ prototypes $c_j$, for $j = 0, 1, \ldots, N_B$, to each block $B_i = \{c_j\}_j$, where $i = 0, 1, \ldots, N_{\mathcal{P}}$.

Instead of mapping a single representation $z_i^v$ as a linear combination of all prototypes in $C$, we compare the view's representations $z_i^v$ and $w_i^v$ against all the prototypes in the $j$-th block. That is, $s_{i,j}^v = q(z_i^v, B_j)$ and $t_{i,j}^v = q(w_i^v, B_j)$, for every block in the partition $\mathcal{P}$, to obtain the normalized probability distribution relating a view from image $i$ with the prototypes of the $j$-th block of the random partition, where $s_{i,j}^v$ and $t_{i,j}^v \in \mathbb{R}^{1 \times 1 \times N_P}$.

To ensure that views are consistent among the blocks, we optimize the views' distributions $s_{i,j}^v$ and $t_{i,j}^v$ over the prototypes of a block indexed by $j$, so that the two distributions are consistent with one another. Thus, the consistency term of our partition loss is $\mathcal{L}_c(s_{i,j}^1, t_{i,j}^2)$, where for each block

$B_j$, the loss is minimized when the pair of views, $\hat{x}_i^1$ and $\hat{x}_i^2$, gets assigned to the same prototypes across blocks. In other words, we look for the agreement between student and teacher assignments' probabilities across views of a given sample.

Following similar reasoning, the block-wise entropy term is defined as $H(\bar{p}_j)$, where $\bar{p}_j = \text{avg}(\{(s_{i,j}^v, t_{i,j}^v)\}_{i=1}^N)$ is the average prediction over each block $B_j$ for a batch of size $N$. Thus, the final objective for consistent assignment of random partition is,

$$\mathcal{L} = \frac{1}{N N_\mathcal{P}} \sum_i^N \sum_j^{N_\mathcal{P}} \left( \mathcal{L}_c(s_{i,j}^1, t_{i,j}^2) + \mathcal{L}_c(s_{i,j}^2, t_{i,j}^1) \right) - H(\bar{p}_j). \tag{4}$$

Note that to fully use the pair of views at each iteration, we symmetrically use the $\mathcal{L}_c$ consistency function. The probability vectors $t_{i,j}^v$ come from the momentum teacher and are used as target distributions.

We can view the random partition pretext task as posing multiple pseudo-classification problems over subsets of prototypes. At each iteration, the pseudo-classification tasks change because the partitions are recreated with different prototypes, cf. Figure 2. One of the benefits of such a strategy is that we no longer require tuning the hyperparameter $\lambda_e$ to avoid trivial solutions. The stochastic nature of the random partition pretext task, blended with the multiple prediction tasks over subsets of prototypes, provides a regularization effect that improves the learned representations and training stability.

Other clustering-based methods [7, 8] rely on sharpening the distributions to improve their self-supervised signals used as targets in a cross-entropy loss. On the contrary, our formulation does not require the temperature parameter for sharpening the predictions and guiding the learning of the student. We can think of the consistency loss as implicitly learning the temperature parameter to make the predictions sharper at each iteration. This is an important advantage of our consistency loss in contrast to previous methods.

## 5 Main results

### 5.1 Transfer learning evaluation

Table 1 shows that CARP's pre-trained representations are suitable for learning new downstream tasks over multiple datasets with distinct difficulty levels. **We compare CARP's $k$-NN performance against nine SSL methods across eight datasets** and report average results for $k= \{10, 20, 100, 200\}$, over all datasets. We advocate for $k$-NN instead of linear evaluation, where a linear classifier is trained on top of the frozen features, for the following reasons: (1) $k$-NN is faster, (2) $k$-NN demands fewer resources, and (3) $k$-NN requires less hyperparameter tuning. For individual references, Table 1 shows top-1 accuracy for $k = 20$ on each dataset. **For fixed $k = 20$, CARP outperforms competitors in 5 out of 8 datasets, and in the Flowers and Country datasets, it is a close second**. Moreover, the average performances at $k$ show that CARP performs comparably well on all datasets without significant performance differences when varying the number of labeled examples $k$. We show the detailed results in Appendix A.1.

### 5.2 Clustering evaluation

Table 2 reports clustering performance metrics of various clustering-based SSL methods on the ImageNet-1M [36], CIFAR-10/100 [27], and the GTSRB [40] datasets. For the ImageNet-1M, only 1% of the labeled data is used following the subset provided by Chen et al. [9]. See Appendix D.4 for the detailed $k$-means evaluation protocol.

### 5.3 Image retrieval and copy detection

Motivated by the strong $k$-NN performance of CARP and following the previous evaluation protocol by Caron et al. [8], we assess the performance of ImageNet pre-trained CARP encoders on image retrieval and copy detection downstream tasks. We took the officially released weights from the competing methods, used the frozen encoders as feature extractors, and performed retrieval using $k$-NN. For both tasks, the nearest neighbor computation is done on the 2048-dim representation from the ResNet-50 encoder. We report the top-3 best-performing methods. For an extended evaluation, see Appendix A.2.

Table 1: **Transfer learning evaluation.** We report top-1 accuracy ($k = 20$) for individual datasets and averages over all datasets for $k \in \{10, 20, 100, 200\}$. Top performing in **bold**, top-2 underlined.

| Methods | Ep | Pets | Flowers | Aircraft | Cars | Country | Food | STL | GTSRB | Avg @k 10 | 20 | 100 | 200 |
|---|---|---|---|---|---|---|---|---|---|---|---|---|---|
| | | results for $k = 20$ | | | | | | | | | | | |
| oBoW (mc) [19] | 200 | 57.3 | 61.9 | 18.1 | 11.5 | 12.0 | 47.4 | 96.6 | 50.6 | 44.3 | 44.4 | 43.5 | 43.0 |
| SeLa-v2 (mc) [1] | 400 | 66.8 | 58.6 | 20.7 | 13.3 | 10.5 | 46.8 | 94.0 | 59.0 | 46.1 | 46.2 | 45.5 | 45.1 |
| InfoMin [41] | 800 | 77.8 | 61.9 | 18.2 | 14.4 | 11.6 | 52.4 | 96.4 | 54.8 | 48.6 | 48.4 | 47.3 | 46.6 |
| DeepC-v2 (mc) [5] | 800 | 78.3 | 76.3 | 32.0 | 25.0 | 13.6 | 62.3 | 95.6 | 63.4 | 56.0 | 55.8 | 54.4 | 53.3 |
| SwAV (mc) [7] | 800 | 77.0 | 75.2 | 29.0 | 22.7 | 13.8 | 59.1 | 95.2 | 63.2 | 54.5 | 54.4 | 53.1 | 52.2 |
| DINO (mc) [8] | 800 | 80.9 | **81.6** | 35.3 | 30.1 | **14.4** | 62.0 | 95.6 | 62.9 | 57.9 | 57.8 | 56.9 | 56.0 |
| Triplet [45] | 980 | 83.5 | 77.7 | 33.4 | 25.2 | 14.1 | 61.5 | 95.6 | 63.5 | 56.5 | 56.8 | 56.2 | 55.6 |
| BarlowT [48] | 1000 | 82.9 | 78.8 | 32.7 | 26.3 | 13.3 | 61.4 | 94.8 | 65.6 | 56.8 | 57.0 | 56.4 | 55.7 |
| MoCo-v3 [12] | 1000 | 86.4 | 79.0 | 36.9 | 29.3 | 12.4 | 60.0 | **96.7** | 72.8 | 59.2 | 59.2 | 58.4 | 57.8 |
| CARP | 400 | **86.8** | 80.0 | **42.1** | **33.5** | 12.3 | 58.4 | 95.9 | **75.3** | **60.4** | **60.5** | **59.7** | **59.2** |
| CARP (mc) | 400 | 83.9 | 80.3 | 34.8 | 27.1 | 14.2 | **62.9** | 95.5 | 62.8 | 57.6 | 57.7 | 56.8 | 56.0 |

Table 2: **Clustering evaluation**. We report (NMI) normalized mutual information, (AMI) adjusted mutual information, and (ARI) adjusted rand index. Top performing in **bold**, top-2 underlined.

| Method | ImageNet-1M NMI | AMI | ARI | CIFAR-10 NMI | AMI | ARI | CIFAR-100 NMI | AMI | ARI | GTSRB NMI | AMI | ARI |
|---|---|---|---|---|---|---|---|---|---|---|---|---|
| PCL v2 [29] | 69.7 | 47.5 | 22.2 | 46.7 | 46.6 | 34.8 | 49.1 | 42.2 | 17.1 | 44.1 | 44.1 | 13.0 |
| SeLa-v2 [1] | 68.7 | 45.5 | 21.3 | 42.0 | 41.9 | 30.6 | 49.7 | 42.9 | 18.2 | 45.7 | 43.2 | 12.0 |
| DeepC-v2 [5] | 69.7 | 47.1 | 22.4 | 47.0 | 46.9 | 35.5 | 53.2 | 46.7 | 21.8 | 48.1 | 45.7 | 13.5 |
| SwAV [7] | 68.5 | 45.1 | 20.5 | 46.8 | 46.8 | 37.0 | 52.1 | 45.5 | 20.0 | 51.0 | 48.8 | 15.0 |
| DINO [8] | 69.2 | 46.2 | 21.7 | 39.6 | 39.5 | 28.0 | 47.6 | 40.4 | 16.2 | 52.0 | 49.8 | 15.4 |
| MIRA [28] | 68.9 | 45.7 | 21.2 | 39.5 | 39.4 | 28.8 | 49.0 | 42.1 | 17.6 | 51.6 | 49.4 | 15.8 |
| CoKe [33] | 68.9 | 45.6 | 21.3 | 45.9 | 45.8 | 34.2 | 51.9 | 45.2 | 19.5 | 49.4 | 47.1 | 13.7 |
| CARP | **70.3** | **48.0** | **23.9** | **49.0** | **48.9** | **38.7** | **54.5** | **48.2** | **23.1** | **54.8** | **52.7** | **19.6** |

**Image retrieval.** We consider the revisited Oxford and Paris landmark image retrieval datasets [34]. Given a query image of a landmark, the objective is to retrieve all database images depicting the same landmark. Each dataset contains three different difficulty levels. In Table 3, we report the mean Average Precision (mAP) on the Medium and Hard subsets for various SSL algorithms. CARP's representations perform well on Oxford 5k and take second place for Paris 6k. See Appendix D.5 for the evaluation protocol.

**Copy detection.** We benchmark self-supervised ResNet-50 encoders on the INRIA Copydays dataset [16]. In Table 4, we report the mean Average Precision (mAP) on the "strong" subset and compare CARP's performance against other state-of-the-art methods.

Table 3: **Image retrieval evaluation.** We report mAP on the revisited Oxford and Paris for the (M) Medium and (H) Hard subsets.

| Method | ep | $\mathcal{R}\mathcal{O}$x M | H | $\mathcal{R}$par M | H |
|---|---|---|---|---|---|
| Supervised [35] | 100 | 49.8 | 8.5 | 74.0 | 52.1 |
| Random | – | 1.6 | 0.7 | 4.1 | 2.5 |
| DINO (mc) [8] | 800 | 35.4 | 11.1 | 55.9 | 27.5 |
| Triplet [45] | 980 | 35.3 | 12.0 | 58.2 | 28.7 |
| MoCo-v3 [12] | 1000 | 33.1 | 10.9 | **59.1** | **31.3** |
| CARP | 200 | **38.8** | **15.5** | 58.8 | 30.4 |

Table 4: **Copy-detection evaluation.** We report mAP for the Copydays dataset on the "strong" subset.

| Method | Ep | mAP |
|---|---|---|
| Random | – | 25.7 |
| DINO (mc) [8] | 800 | 78.8 |
| Triplet [45] | 980 | 81.7 |
| VICReg [3] | 1000 | 83.7 |
| MoCo-v3 [12] | 1000 | 80.6 |
| CARP (mc) | 400 | **84.0** |

Table 5: **Few-shot classification on VOC07 and INat2018**. We report mAP for VOC07 and top-1 accuracy for INat2018, at $n$, across 5 independent runs, where $n$ denotes the number of training examples. Top performing in **bold**, top-2 underlined.

| Method | Ep | Pascal VOC07 | | | | | | INat2018 | | | | | |
|---|---|---|---|---|---|---|---|---|---|---|---|---|---|
| | | n=1 | n=2 | n=4 | n=8 | n=16 | full | n=1 | n=2 | n=4 | n=8 | n=16 | full |
| PCL v2 [29] | 200 | **47.9** | 59.6 | 66.2 | 74.5 | 78.3 | 85.4 | 1.4 | 1.6 | 2.3 | 2.9 | 4.8 | 2.1 |
| DINO (mc) [8] | 800 | 45.6 | 58.4 | 66.6 | 74.8 | 79.6 | 88.2 | 6.5 | 12.0 | 20.4 | 29.6 | 35.9 | 30.4 |
| Triplet [45] | 980 | 43.6 | 56.2 | 64.6 | 73.8 | 79.6 | **88.3** | 11.4 | 19.1 | 28.9 | 37.6 | 44.0 | 41.4 |
| MoCo-v3 [12] | 1000 | 46.6 | 59.6 | 67.0 | 75.4 | **80.2** | 87.4 | 8.1 | 12.2 | 18.5 | 27.2 | 33.5 | 28.0 |
| CARP (mc) | 200 | 46.0 | 58.3 | 66.5 | 75.5 | 79.5 | 88.0 | 8.6 | 14.4 | 23.6 | 32.7 | 38.2 | 33.9 |
| | 400 | 47.1 | **59.8** | **67.3** | 75.8 | 80.0 | 88.2 | **11.5** | **19.6** | **29.6** | **39.1** | **45.1** | **42.6** |

## 5.4 Few-shot classification

Table 5 compares the few-shot classification performance of SSL methods on the VOC07 [17] and INat2018 [43] datasets. We train linear SVMs following Li et al. [29] and linear classifiers on top of the frozen representations from the self-supervised ResNet-50 encoders for VOC07 and INat2018, respectively.

The INat2018 dataset is especially challenging for low-shot classification. It contains 8142 classes with a long tail distribution, where the number of images per class varies between 1000 maximum and 2 minimum. **CARP remains a strong feature extractor for both tasks while competitors oscillate between datasets.** CARP pre-trained for 400 epochs demonstrates an efficient learning performance and wins most configurations. We report a complete evaluation with standard deviations in Appendix A.3.

## 5.5 Linear evaluation

Following the linear evaluation protocol proposed by Zhou et al. [49], we trained linear classifiers on top of CARP's frozen 2048-dim representations for 100 epochs.

We assess the linear evaluation performance of CARP's pre-trained representations on ImageNet-1M for three pre-training configurations, 100, 200, and 400 epochs, varying the utilization of multi-crop (mc) augmentation, Table 6. CARP surpasses SwAV on all pre-trained configurations, +0.4% on 100 epochs, +0.4% on 200 epochs, +0.4% on 400 epochs w/o multi-crop, and +0.4% on 400 epochs with multi-crop. Indeed, CARP's 400 epochs pre-trained representations perform on par with DINO and SwAV, both pre-trained for 800 epochs.

In addition, we evaluated CARP's representations using a weighted $k$-Nearest Neighbor ($k$-NN) classifier. CARP achieves better $k$-NN performance than SwAV (+1.4%), DINO (+0.2%), and BYOL (+1.1%), all trained for 800 epochs or more. These results emphasize the efficiency of the proposed random partition pretext task.

## 6 Ablations

In this section, we assess whether a consistent assignment of views over random partitions benefits the learned representations and improves training stability. We ablate CARP's main hyperparameters to establish a good baseline for pre-training on ImageNet-1M. For ablations, we trained CARP using the full ImageNet-1M dataset for 50 epochs. The batch size was set to 256, the number of prototypes $K = 65\,536$, and the number of random partition blocks $N_{\mathcal{P}} = 128$. Hence, each block contains $N_B = 512$ prototypes. We report results for single runs. See Appendix B for more results and Appendix C for implementation details.

### 6.1 Training CARP with different batch sizes

Most SSL methods [7, 8, 22, 48] report their best results when using substantially large batch sizes. In Table 7, we observe a similar pattern when training CARP. Our default configuration of 1024

Table 6: **Linear evaluation**. We report top-1 linear and $k$-NN ($k = 20$) accuracy for the ImageNet-1M dataset. [†] Results computed by us using the officially released pre-trained models. Top performing in **bold**, top-2 underlined.

| Method | Ep | Linear | $k$-NN |
|---|---|---|---|
| Supervised | 100 | 76.5 | – |
| SwAV (mc) [7] | 100 | 72.1 | 61.8[†] |
| | 200 | 73.9 | 63.7[†] |
| | 400 | 74.6 | 65.0[†] |
| | 800 | **75.3** | 66.3[†] |
| DINO (mc) [8] | 800 | **75.3** | 67.5 |
| BYOL [22] | 1000 | 74.3 | 66.6 |
| MoCo-v3 [24] | 1000 | 74.6 | **68.9**[†] |
| CARP | 400 | 73.0 | 67.6 |
| CARP (mc) | 100 | 72.5 | 63.5 |
| | 200 | 74.2 | 66.5 |
| | 400 | **75.3** | 67.7 |

Table 7: CARP learns better representations when larger batch sizes (bs) are employed.

| bs | 128 | 256 | 512 | 1024 | 2048 | 4096 |
|---|---|---|---|---|---|---|
| $k$-NN | 46.56 | 51.32 | 54.23 | 56.63 | 57.0 | **58.5** |

Table 8: Exploring different strategies to create the partitions.

| Epochs | 25 | 50 | 75 | 100 |
|---|---|---|---|---|
| Constant | 45.15 | 49.89 | 52.81 | 53.32 |
| Random | **48.68** | **53.98** | **55.93** | **56.38** |

observations yields a $k$-NN top-1 performance 10% higher than a batch size of 128. Table 7 confirms that training with large batch sizes benefits the learned representations. However, training with smaller batch sizes requires further tuning of other hyperparameters, such as the block size $N_B$. Specifically, we observed that reducing the block size $N_B$ improves the learned representations when training with small batch sizes, which makes CARP robust to low-resource training.

## 6.2 Exploring different strategies to build partitions

Table 8 explores different ways of creating random partition blocks from the learnable prototypes. CARP's default strategy recreates the random partitions at every training step. In other words, for each iteration, we assign $N_B$ randomly chosen prototypes to the $N_\mathcal{P}$ partition blocks. Table 8 contrasts CARP's default strategy with one in which the partition blocks are created only once, in a sequential manner, and kept fixed throughout training. We observe that training CARP with fixed partition blocks still produce useful representations. However, as measured by $k$-NN performance, randomly recreating the partition blocks at each iteration further benefits the learned representations. Since the partition blocks are randomly recreated at every iteration of gradient descent, the classification subproblems are always different. In practice, this variance allows for many unique pretext tasks at each iteration, which provides a positive regularization effect on CARP.

## 7  Limitations

Even though CARP's representations transfer well to many downstream tasks, our experiments showed that CARP's representations do not transfer well to dense prediction tasks such as detection and segmentation. We hypothesize this limitation is due to the architectural characteristics of ConvNets that collapse local feature maps to an average global representation in the last layer, combined with our classification-like loss function. Refer to Appendix A.4 for quantitative results.

# 8 Conclusion

We presented consistent assignment of views over random partitions (CARP), a self-supervised clustering-based method for visual feature learning. CARP learns prototypes in an online fashion end-to-end using gradient descent by minimizing a cost function that optimizes consistency between views' assignments and uniform distribution across prototypes within a random partition. Our experiments demonstrate that posing the optimization problem at the level of random partitions of learnable prototypes stabilizes training by avoiding trivial solutions in joint-embedding architectures and increases the performance of the learned representation. We compared the performance of CARP against the state-of-the-art ResNet-based SSL methods across multiple pretext tasks and datasets. The results demonstrated that the representations learned by CARP performed well on many visual downstream tasks.

## Acknowledgements

The computations were performed in part on resources provided by Sigma2—the National Infrastructure for High Performance Computing and Data Storage in Norway—through Project NN8104K. This work was funded in part by the Research Council of Norway, through its Centre for Research-based Innovation funding scheme (grant no. 309439), and Consortium Partners. This study was financed in part by the Coordenação de Aperfeiçoamento de Pessoal de Nível Superior—Brasil (CAPES)—Finance Code 001

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

# A  Extended results

In this section, we expand the evaluation experiments from the main text. We use the officially released pre-trained methods across all benchmarks to represent each SSL method. We did not retrain/reimplement any competing method. In total, we pre-trained 4 instances of CARP. To evaluate CARP's performance in a low training regime, we trained two 200-epoch models, one with multi-crop (mc) and the other without it. Similarly, to evaluate longer training performance, we trained two 400-epoch models, one with multi-crop and the other without it.

## A.1  Transfer learning evaluation

In Table A.1, we report detailed results for the transfer learning $k$-NN experiments. We evaluate SSL methods for values of $k \in \{10, 20, 100, 200\}$, including all instances of CARP. Cf. Appendix D.3 for the evaluation protocol.

**CARP achieved either top-1 or top-2 performance in seven out of 8 datasets.** STL-10 is the only dataset where CARP is neither the top-1 nor top-2. **Moreover, in the FGVCAircraft, Stanford Cars, and GTSRB datasets, CARP archived top-1 and top-2 performances with large margins.** The average $k$-NN performance per value of $k$ across all datasets is reported in Table 1 in the main text.

## A.2  Image retrieval and copy detection

Tables A.2 and A.3 show additional results for image retrieval and copy detection evaluations. For both tasks, we compare CARP's performance against nine SSL methods. We report mAP on the Medium and Hard splits of the revisited Oxford and Paris datasets for image retrieval. We report mAP on the "strong" set of the Copydays dataset for copy detection.

## A.3  Few-shot evaluation

Table A.4 shows additional few-shot evaluation results on the Pascal VOC07 and INat-2018 datasets. We report mAP and top-1 accuracy @$k$, averaged over 5 independent runs, for VOC07 and INat-2018, respectively. We include CARP's 200 and 400 multi-crop models. Cf. Appendix D.2 for details on the evaluation protocol.

## A.4  Dense prediction evaluation

We evaluated our CARP's multi-crop model on the object detection downstream tasks using the Pascal VOC07 dataset. We followed the guidelines from He et al. [24]. We fine-tuned for 24k iterations on PASCAL VOC trainval07+12 and evaluated on test2007. CARP performs at AP=44.2, AP50=79.4, and AP75=47.6; results are averaged over 5 trials. Compared to other SSL methods, such as MoCo-v2 [11] (AP=57.4 AP50=82.5 AP75=64.0), CARP underperforms significantly.

# B  Ablations

Due to a limited execution budget, the ablations and the main experiments differ slightly in some hyperparameters. Here, we describe the configurations used for the ablations—for the main experiments, see Appendix C.1. For ablations, we trained CARP using the full ImageNet-1M dataset for 50 epochs. The projection head learns a latent representation of 128-dim. The batch size is set to 256 observations, and the projection head hidden layers contain 2048 neurons. We set the number of learnable prototypes $K = 65\,536$, and the number of random partition blocks $N_{\mathcal{P}} = 128$. Hence, each block contains $N_B = 512$ prototypes. We report results for single runs.

## B.1  Does the number of learnable prototypes affect the learned representations?

Table B.1 examines the effect of training CARP with different configurations of prototypes $K$. Similar to other clustering-based SSL methods [7, 8, 29], CARP also benefits from over-clustering. As the number of trainable prototypes $K$ grows, the $k$-NN performance of the learned representations increases. In addition, note that if the number of prototypes $K$ is smaller than the number of actual

Table A.1: **Transfer learning evaluation**. We compare CARP's performance against nine SSL methods on eight datasets. We report results for $k \in \{10, 20, 100, 200\}$. Top methods in **bold**, top-2 underlined. Methods with (mc) use multi-crop.

| Method | ep | Oxford-IIIT Pet | | | | Oxford Flowers-102 | | | | FGVCAircraft | | | | Stanford Cars | | | |
|---|---|---|---|---|---|---|---|---|---|---|---|---|---|---|---|---|---|
| | | 10 | 20 | 100 | 200 | 10 | 20 | 100 | 200 | 10 | 20 | 100 | 200 | 10 | 20 | 100 | 200 |
| oBoW (mc) [18] | 200 | 55.8 | 57.3 | 57.2 | 57.5 | 63.5 | 61.9 | 58.9 | 59.4 | 19.1 | 18.1 | 15.8 | 14.9 | 11.9 | 11.5 | 10.9 | 10.5 |
| SeLa-v2 [1] | 400 | 66.7 | 66.8 | 65.8 | 66.0 | 59.5 | 58.6 | 56.8 | 57.2 | 21.3 | 20.7 | 18.1 | 16.4 | 13.3 | 13.3 | 13.5 | 13.4 |
| InfoMin [41] | 800 | 78.0 | 77.8 | 76.6 | 76.2 | 63.6 | 61.9 | 60.1 | 60.8 | 18.9 | 18.2 | 15.8 | 13.4 | 14.7 | 14.4 | 13.2 | 12.9 |
| SWAV (mc) [7] | 800 | 77.2 | 77.0 | 74.9 | 74.5 | 76.4 | 75.2 | 73.7 | 74.6 | 29.6 | 29.0 | 27.5 | 25.1 | 22.7 | 22.7 | 21.8 | 21.0 |
| DINO (mc) [8] | 800 | 81.5 | 80.9 | 79.0 | 78.9 | **82.3** | **81.6** | **80.8** | **81.2** | 36.1 | 35.3 | 33.5 | 31.1 | 30.0 | 30.1 | 28.9 | 27.5 |
| DeepC-v2 (mc) [5] | 800 | 79.0 | 78.3 | 76.3 | 75.4 | 78.3 | 76.3 | 75.3 | 76.0 | 32.5 | 32.0 | 28.9 | 26.5 | 25.2 | 25.0 | 23.4 | 22.1 |
| Triplet [45] | 980 | 83.3 | 83.5 | 82.4 | 82.4 | 78.5 | 77.7 | 76.9 | 77.3 | 33.3 | 33.4 | 31.7 | 29.6 | 24.4 | 25.2 | 25.5 | 25.0 |
| MoCo-v3 [12] | 1000 | 86.6 | 86.4 | 85.8 | 85.8 | 79.8 | 79.0 | 78.3 | 78.6 | 37.7 | 36.9 | 33.5 | 32.1 | 28.6 | 29.3 | 28.4 | 27.2 |
| BarlowT [48] | 1000 | 82.5 | 82.9 | 82.2 | 82.3 | 79.8 | 78.8 | 77.9 | 78.1 | 32.9 | 32.7 | 30.6 | 29.2 | 25.9 | 26.3 | 26.1 | 25.2 |
| CARP | 200 | **86.8** | **86.8** | **86.2** | 85.9 | 79.0 | 78.2 | 77.4 | 77.7 | 40.5 | 38.9 | 36.0 | 34.4 | 29.4 | 29.8 | 29.6 | 29.6 |
| | 400 | 86.4 | **86.8** | **86.2** | **86.0** | 81.0 | 80.0 | 79.3 | 79.6 | **42.5** | **42.1** | **39.3** | **38.6** | **32.6** | **33.5** | **32.6** | **31.5** |
| CARP (mc) | 200 | 78.7 | 78.7 | 77.1 | 76.8 | 80.6 | 79.7 | 78.7 | 78.8 | 35.7 | 35.0 | 32.4 | 30.6 | 26.4 | 26.6 | 25.4 | 24.3 |
| | 400 | 83.9 | 83.9 | 83.6 | 83.2 | 81.4 | 80.3 | 79.0 | 79.6 | 35.2 | 34.8 | 32.4 | 30.7 | 26.6 | 27.1 | 26.1 | 24.9 |

| Method | ep | Country-211 | | | | Food-101 | | | | STL-10 | | | | GTSRB | | | |
|---|---|---|---|---|---|---|---|---|---|---|---|---|---|---|---|---|---|
| | | 10 | 20 | 100 | 200 | 10 | 20 | 100 | 200 | 10 | 20 | 100 | 200 | 10 | 20 | 100 | 200 |
| oBoW (mc) [18] | 200 | 11.7 | 12.0 | 11.8 | 11.4 | 45.8 | 47.4 | 47.3 | 46.0 | 96.6 | 96.6 | **96.3** | 95.7 | 50.1 | 50.6 | 49.9 | 48.2 |
| SeLa-v2 [1] | 400 | 10.1 | 10.5 | 11.0 | 11.0 | 45.7 | 46.8 | 46.6 | 45.5 | 94.0 | 94.0 | 93.5 | 93.3 | 58.1 | 59.0 | 58.7 | 57.9 |
| InfoMin [41] | 800 | 11.2 | 11.6 | 11.9 | 11.9 | 51.5 | 52.4 | 51.4 | 49.5 | 96.5 | 96.4 | 96.2 | **96.0** | 54.9 | 54.8 | 53.5 | 52.3 |
| SwAV (mc) [7] | 800 | 13.6 | 13.8 | 13.1 | 12.7 | 57.9 | 59.1 | 58.2 | 57.0 | 95.5 | 95.2 | 94.0 | 93.0 | 62.9 | 63.2 | 61.6 | 60.0 |
| DINO (mc) [8] | 800 | 14.1 | 14.4 | 14.2 | 13.6 | 60.9 | 62.0 | 61.4 | 60.0 | 95.9 | 95.6 | 94.7 | 93.8 | 62.7 | 62.9 | 62.6 | 61.8 |
| DeepC-v2 (mc) [5] | 800 | 13.3 | 13.6 | 12.5 | 12.1 | 61.2 | 62.3 | 61.4 | 60.1 | 95.7 | 95.6 | 94.5 | 93.3 | 62.9 | 63.4 | 62.4 | 61.3 |
| Triplet [45] | 980 | 13.7 | 14.1 | **14.3** | **14.2** | 60.1 | 61.5 | 61.0 | 60.0 | 95.7 | 95.6 | 94.9 | 94.5 | 63.4 | 63.5 | 62.9 | 62.0 |
| MoCo-v3 [12] | 1000 | 12.4 | 12.4 | 13.2 | 13.2 | 59.0 | 60.0 | 59.1 | 57.7 | **96.9** | **96.7** | 96.2 | 95.9 | 72.4 | 72.8 | 72.6 | 71.7 |
| BarlowT [48] | 1000 | 12.8 | 13.3 | 13.7 | 13.6 | 60.3 | 61.4 | 60.7 | 59.4 | 94.8 | 94.8 | 94.2 | 93.8 | 65.3 | 65.6 | 65.6 | 64.4 |
| CARP | 200 | 11.9 | 12.2 | 12.7 | 12.8 | 57.4 | 58.4 | 57.7 | 56.3 | 95.5 | 95.5 | 94.6 | 93.9 | 73.1 | 73.7 | 73.5 | 72.6 |
| | 400 | 11.9 | 12.3 | 12.8 | 12.8 | 57.6 | 58.4 | 57.6 | 56.3 | 96.1 | 95.9 | 95.0 | 94.3 | **74.7** | **75.3** | **75.2** | **74.4** |
| CARP (mc) | 200 | **14.2** | **14.5** | **14.3** | 13.9 | 60.5 | 61.8 | 60.7 | 59.3 | 95.8 | 95.5 | 94.1 | 93.4 | 64.6 | 64.7 | 64.2 | 63.0 |
| | 400 | 14.1 | 14.2 | **14.3** | 13.9 | **61.7** | **62.9** | **62.1** | **60.7** | 95.9 | 95.5 | 94.3 | 93.4 | 62.2 | 62.8 | 62.4 | 61.4 |

classes in the dataset, the $k$-NN performance of the learned representations degrades. Based on these experiments, we set the default number of prototypes $K = 65536$ for the ImageNet-1M dataset.

## B.2 Does the number of partition blocks matter?

To better understand the effect of the hyperparameters $N_{\mathcal{P}}$ and $N_B$ on the learned representations and in the training stability, the first row of Table B.2 demonstrates the performance of CARP using different configurations for the number of partition blocks $N_{\mathcal{P}}$ and their sizes $N_B$. For completeness, we analyze the effect of removing the momentum encoder in Appendix B.3. We also present an ablation on the effect of the momentum update in Table B.3.

Specifically, as the partition sizes grow and the number of partition blocks $N_{\mathcal{P}}$ decreases, the quality of the learned representations tends to decline and eventually collapse. Note that setting a partition size $N_B = 65536$ produces a single partition block $N_{\mathcal{P}}$ containing all prototypes. Precisely, the setup in the last row and last column of Table B.2 is equivalent to CARL [38]. It shows that a naive implementation leads to a collapsed solution, and the divide-and-conquer approach of devising random partitions from the learnable prototypes avoids such trivialities.

Note that as smaller the block size $N_B$, more stable the algorithm will be. However, the quality of the learned representation might decrease since the pseudo-classification tasks, posed by the random partitions, becomes easier with fewer prototypes. On the other hand, a larger block size $N_B$ poses a more challenging consistency task at the expense of contributing to mode collapse.

For most cases, however, for block sizes ranging from $N_B = 128$ to $N_B = 4096$, CARP learns useful representations and shows robustness to this hyperparameter. By default we set the partition block size $N_B = 512$.

Table A.2: **Image retrieval evaluation.** We report mAP performance of various self-supervised methods for the image retrieval downstream task on the revisited Oxford and Paris datasets. All SLL methods were pre-trained on ImageNet-1M. We used the officially released pre-trained models from respective methods for evaluation. Top-1 performers in **bold**, top-2 underlined.

| Method | Ep | $\mathcal{RO}$x | | $\mathcal{R}$par | |
| | | Medium | Hard | Medium | Hard |
| --- | --- | --- | --- | --- | --- |
| Supervised | 100 | 49.8 | 18.5 | 74.0 | 52.1 |
| Scratch | | 1.6 | 0.7 | 4.1 | 2.5 |
| oBoW (mc) [18] | 200 | 20.4 | 4.4 | 40.6 | 16.2 |
| SeLa-v2 (mc) [1] | 400 | 20.1 | 4.9 | 37.1 | 13.6 |
| InfoMin [41] | 800 | 24.4 | 5.7 | 44.6 | 18.8 |
| SwAV (mc) [7] | 800 | 31.1 | 10.1 | 48.9 | 20.6 |
| DINO (mc) [8] | 800 | 35.4 | 11.1 | 55.9 | 27.5 |
| DeepC-v2 (mc) [5] | 800 | 32.6 | 10.9 | 50.0 | 20.2 |
| Triplet [45] | 980 | 35.3 | 12.0 | 58.2 | 28.7 |
| VICReg [3] | 1000 | 32.7 | 8.5 | 57.5 | 29.0 |
| MoCo-v3 [12] | 1000 | 33.1 | 10.9 | **59.1** | **31.3** |
| CARP | 200 | 38.8 | **15.5** | 58.8 | 30.4 |
| | 400 | **38.9** | 15.1 | 58.5 | 30.2 |
| CARP (mc) | 200 | 32.8 | 10.4 | 53.6 | 24.9 |
| | 400 | 33.7 | 11.6 | 54.0 | 26.5 |

Table A.3: **Copy detection evaluation**. We report mAP on the "strong" subset of the Copydays dataset and compare CARP's performance against seven SSL methods. Top-1 performers in **bold**, top-2 underlined.

| Method | Ep | mAP |
| --- | --- | --- |
| Scratch | | 25.7 |
| oBoW (mc) [18] | 200 | 61.5 |
| SeLa-v2 (mc) [1] | 400 | 76.6 |
| InfoMin [41] | 800 | 67.5 |
| SwAV (mc) [7] | 800 | 76.1 |
| DINO (mc) [8] | 800 | 78.8 |
| DeepC-v2 (mc) [5] | 800 | 76.0 |
| Triplet [45] | 980 | 81.7 |
| VICReg [3] | 1000 | 83.7 |
| MoCo-v3 [12] | 1000 | 80.6 |
| CARP | 200 | 82.3 |
| | 400 | 82.6 |
| CARP (mc) | 200 | 80.8 |
| | 400 | **84.0** |

## B.3 The importance of the momentum encoder

Table B.2 contrasts CARP's joint-embedding architectures with and without a momentum encoder, which is equivalent to setting $\eta = 0$ in the momentum encoder update equation. Different from other SSL methods [8, 22], CARP works with both setups. However, we observe that using a momentum encoder significantly boosts the performance of the learned representations. Table B.2 shows that regardless of block sizes, representations learned using a momentum encoder-based architecture consistently outperform the siamese counterpart.

## B.4 Who provides the best features for downstream evaluation?

One way to understand CARP's joint-embedding architecture with a momentum encoder is through the teacher-student framework, where the momentum encoder is the teacher that guides the learning student. The addition of the momentum encoder raises the question of which module produces the

Table A.4: **Few-shot classification on VOC07 and INat-2018**. We report mAP at $n$ on VOC07 and top-1 accuracy for INat-2018 across 5 independent runs, where $n$ denotes the number of training examples. Standard deviations rounded to the first decimal place.

| | | Pascal VOC07 | | | | | |
|---|---|---|---|---|---|---|---|
| Method | Ep | n=1 | n=2 | n=4 | n=8 | n=16 | full |
| PCL v2 [29] | 200 | **47.9 ± 4.1** | 59.6 ± 2.7 | 66.2 ± 2.2 | 74.5 ± 0.5 | 78.3 ± 0.4 | 85.4 |
| SeLa-v2 (mc) [1] | 400 | 42.0 ± 2.2 | 54.5 ± 3.2 | 62.2 ± 1.5 | 71.4 ± 0.5 | 76.9 ± 0.4 | 85.3 |
| DeepC-v2 (mc) [5] | 800 | 46.5 ± 2.4 | 58.4 ± 2.9 | 66.5 ± 1.6 | 74.5 ± 0.9 | 79.5 ± 0.4 | 87.6 |
| SwAV (mc) [7] | 800 | 42.9 ± 2.1 | 54.9 ± 4.4 | 64.0 ± 2.1 | 72.9 ± 1.1 | 78.7 ± 0.6 | 88.1 |
| DINO (mc) [8] | 800 | 45.6 ± 2.4 | 58.4 ± 3.2 | 66.6 ± 2.1 | 74.8 ± 0.8 | 79.6 ± 0.6 | 88.2 |
| Triplet [45] | 980 | 43.6 ± 3.3 | 56.2 ± 3.5 | 64.6 ± 1.8 | 73.8 ± 0.1 | 79.6 ± 0.7 | **88.3** |
| MoCo-v3 [12] | 1000 | 46.6 ± 3.7 | 59.6 ± 2.9 | 67.0 ± 2.4 | 75.4 ± 0.7 | **80.2 ± 0.6** | 87.4 |
| BarlowT [48] | 1000 | 42.6 ± 3.7 | 55.5 ± 3.2 | 63.5 ± 1.8 | 72.6 ± 0.1 | 77.6 ± 0.5 | 86.3 |
| CARP (mc) | 200 | 46.0 ± 3.2 | 58.3 ± 3.3 | 66.5 ± 2.4 | 75.5 ± 0.1 | 79.5 ± 0.6 | 88.0 |
| | 400 | 47.1 ± 3.2 | **59.8 ± 3.2** | **67.3 ± 2.2** | **75.8 ± 1.1** | 80.0 ± 0.7 | 88.2 |

| | | INat-2018 | | | | | |
|---|---|---|---|---|---|---|---|
| Method | Ep | n=1 | n=2 | n=4 | n=8 | n=16 | full |
| PCL [29] | 200 | 1.4 ± 0.1 | 1.6 ± 0.1 | 2.3 ± 0.2 | 2.9 ± 0.1 | 4.8 ± 0.1 | 2.1 |
| SeLa-v2 (mc) [1] | 400 | 2.9 ± 0.2 | 4.2 ± 0.1 | 6.3 ± 0.1 | 10.0 ± 0.1 | 13.5 ± 0.1 | 8.2 |
| DeepC-v2 (mc) [5] | 800 | 7.6 ± 0.2 | 13.0 ± 0.8 | 20.9 ± 0.5 | 29.6 ± 0.4 | 36.4 ± 0.2 | 32.8 |
| SwAV (mc) [7] | 800 | 5.3 ± 0.1 | 9.2 ± 0.5 | 15.6 ± 0.1 | 23.1 ± 0.2 | 29.4 ± 0.2 | 24.2 |
| DINO [8] | 800 | 6.5 ± 0.1 | 12.0 ± 0.5 | 20.4 ± 0.5 | 29.6 ± 0.3 | 35.9 ± 0.3 | 30.4 |
| Triplet [45] | 980 | 11.4 ± 0.2 | 19.1 ± 0.7 | 28.9 ± 0.8 | 37.6 ± 0.3 | 44.0 ± 0.1 | 41.4 |
| MoCo-v3 [12] | 1000 | 8.1 ± 0.1 | 12.2 ± 0.3 | 18.5 ± 0.3 | 27.2 ± 0.3 | 33.5 ± 0.1 | 28.0 |
| BarlowT [48] | 1000 | 8.8 ± 0.1 | 12.2 ± 0.5 | 17.2 ± 0.2 | 24.6 ± 0.1 | 30.8 ± 0.1 | 25.3 |
| CARP (mc) | 200 | 8.6 ± 0.2 | 14.4 ± 0.1 | 23.6 ± 0.3 | 32.7 ± 0.3 | 38.2 ± 0.2 | 33.9 |
| | 400 | **11.5 ± 0.1** | **19.6 ± 0.1** | **29.6 ± 0.9** | **39.1 ± 0.3** | **45.1 ± 0.2** | **42.6** |

Table B.1: CARP benefits from over-clustering. Setting a small number of prototypes may hurt the learned representations.

| $K$ | 1024 | 2048 | 4096 | 16384 | 65536 | 262144 |
|---|---|---|---|---|---|---|
| $k$-NN | 48.81 | 49.98 | 50.69 | 50.81 | 51.2 | **51.31** |

best representations. To answer this question, Figure B.1 explores the $k$-NN performance when extracting features from the momentum encoder (teacher) versus the student. We observe that teachers' representations constantly outperform the students' during training. However, by the end of the training, the student catches up with the teacher.

## C   Implementation Details

### C.1   Experimental Setup

We train CARP on the ImageNet-1M unlabeled dataset using ResNet50 [23] encoders. We take the output representation of the last global average pooling layer (a 2048-dim vector) and project it to a 256-dim vector. Following Caron et al.'s [8] work, our MLP projection head contains 3 dense layers with batch normalization and the GELU activations. The hidden units of the projection head contain 2048 neurons. The 256-dim representation vector is fed to an assigner MLP that outputs unnormalized probabilities w.r.t. the learnable prototypes. By default, the assigner function is implemented as a linear layer and trains $K = 65\,536$ prototypes. To generate the random partitions, we set the number of partitions $N_{\mathcal{P}} = 128$, which creates subsets containing $N_B = 512$ randomly chosen prototypes. We use the same data augmentations proposed by Grill et al. [22] to generate synthetic views. The protocol creates three data augmentation pipelines, the first two to generate global views and the last to generate multi-crops. CARP is pre-trained with the LARS [47] optimizer, end to end, with weight decay of $1 \times 10^{-6}$. For models training up to 200 epochs, the learning rate starts from 0.6 and decays

Table B.2: CARP with and without a momentum encoder. Without the random partition strategy (last column), training collapses regardless of using a momentum encoder or a pure siamese architecture.

| $N_B$ | 32 | 64 | 128 | 256 | 512 | 1024 | 2048 | 4096 | 16384 | 65536 |
|---|---|---|---|---|---|---|---|---|---|---|
| w/ mom. enc. | 49.56 | 50.75 | 51.19 | 51.20 | 51.32 | 51.06 | 51.31 | 51.08 | 49.67 | 0.11 |
| w/o mom. enc. | 48.95 | 49.28 | 48.81 | 47.37 | 46.16 | 44.68 | 44.29 | 44.39 | 47.25 | 0.11 |

Table B.3: The effect of the hyperparameter $\eta$ on the momentum encoder updates. In the last column, $\eta$ starts as $\eta = 0.99$ and it is annealed to $\eta = 1.0$ following a cosine schedule.

| $\eta$ | 0 | 0.5 | 0.9 | 0.99 | 0.999 | $0.99 \to 1.0$ |
|---|---|---|---|---|---|---|
| $k$-NN | 51.0 | 50.2 | 50.3 | 51.1 | 50.1 | **51.3** |

to 0.006 with a cosine scheduling [30] without warmups. For models pre-trained for more than 400 epochs, the learning rate starts at 0.3 and decays to 0.003 using the same cosine scheduler. We train the system with a global batch size of 4096 observations. For all experiments, we used 4 A100 40GB GPUs and gradient accumulation to simulate large batch sizes. Cf. to Appendix E for a PyTorch style pseudo-code.

# D  Evaluation Protocols

## D.1  Linear evaluation

For ImageNet-1M evaluation, we trained a linear classifier on top of the frozen representations extracted from the last average pooling layer of the ResNet50 encoder for 100 epochs, following Zhou et al.'s [49] protocol. The evaluation script performs a grid search hyperparameter tuning on the learning rate, weight decay regularization, and optimizer. For each input image, we take a random crop followed by a resize to $224 \times 224$ and an optional horizontal flipping. For testing, images are resized to $256 \times 256$ and center-cropped to $224 \times 224$.

## D.2  Few-shot evaluation

We measure the few-shot learning capabilities of SSL methods on the Pascal VOC07 and iNaturalist 2018 datasets. For VOC07, we are interested in the multi-label classification performance. We closely follow Li et al.'s [29] protocol and train Linear SVNs on fixed 2048-dim representations from many SSL ResNet-50 encoders.

For INat-2018, we expand the few-shot evaluation challenge to a complex scenario containing more than 8k classes. We train linear classifiers for 20 epochs on fixed 2048-dim representations. We use

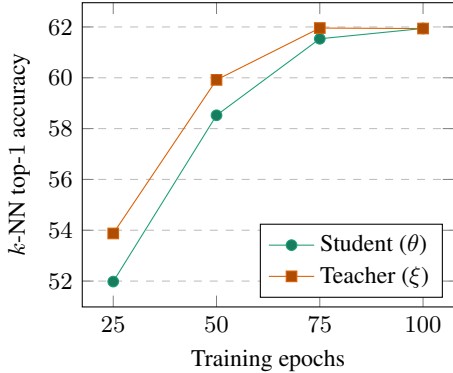

Figure B.1: During training, the representations extracted from the teacher outperform the representations from the student network.

the SGD optimizer. The learning rate starts at 0.03 and decays by a factor of 10 in epochs 12 and 18, respectively.

For both datasets, we vary the number $n$ of labeled examples per class and report the average results across 5 independent runs.

### D.3 $k$-NN evaluation

To perform the $k$-NN evaluation, we use pre-trained SSL ResNet50 encoders as feature extractors to compute and store the representations from many vision datasets. Following Caron et al.'s [8] setup, the representation vector for a test image is compared against all representations from the training split and a prediction is made via weighted voting. If one of the closest neighbors has the same class as the test image, it contributes to the final voting as $\alpha_i = \exp\left(\frac{M_i z}{\tau}\right)$ where $M$ is a memory bank containing representations from the training data, $z$ is the representation from the test data, and $\tau$ is the temperature hyper-parameter. For all experiments, we run $k$-NN with configurations of $K_{\text{near}} \in \{10, 20, 100, 200\}$. For most experiments, a value of $k = 20$ is consistently the best setup across all methods.

### D.4 $k$-means evaluation

Similar to $k$-NN evaluation, we take self-supervised pre-trained encoders and use them to extract 2048-dim feature vectors from the training set of datasets like CIFAR-10/100 and ImageNet-1M. For ImageNet-1M, we use only 10% of the training data following the 10% subset from Chen et al. [9]. We fit $k$-means classifiers on the learned representations of the training set and use the validation split to assess the quality of the learned prototypes. We report three metrics to assess clustering performance: Normalized Mutual Information (NMI), Adjusted Mutual Information (AMI), and Adjusted Rand Index (ARI). The number of prototypes $k$ is set to be the number of true classes of each dataset. We use the faiss library [25] for fast $k$-means. For each experiment, we run $k$-means for 100 iterations, 20 redos, and spherical normalization. To measure the clustering performance of CARP, we observed that a 400 epoch model with a learning rate of 0.3 slightly outperformed the other instances; therefore, we use this model to report results in Table 2. This instance of CARP uses two views and is only used for clustering evaluation.

### D.5 Image retrieval evaluation

We strictly follow the evaluation script `eval_image_retrieval.py` provided by Caron et al. [8], for the image retrieval evaluation task. The script uses the revisited Oxford and Paris image retrieval datasets [34]. The dataset contains three protocols of varying difficulty levels. We take the ImageNet pre-trained ResNet-50 encoder from CARP, freeze the weights, and apply $k$-NN evaluation directly to the frozen 2048-d features for retrieval, conditioned on a query image.

### D.6 Copy detection evaluation

We strictly follow the evaluation script `eval_copy_detection.py` provided by Caron et al. [8] for copy detection evaluation. The evaluation is performed on the INRIA Copydays dataset [16]. The dataset contains holiday pictures in the format query/database. Each image has suffered three kinds of artificial attacks: JPEG, cropping, and "strong." We report performance evaluation on the "strong" subset. Images in the "strong" subset were intentionally distorted by blur, insertions, print, and scan. The task is to recognize these images despite distortion. We take the frozen CARP ResNet-50 encoder and extract 2048-dim vectors from query and database images at resolution $320^2$. Then, we perform copy detection with cosine similarity between query and database features. We report mean average precision (mAP) as a performance metric. Unlike the benchmark described by Caron et al. [8], we do not utilize additional distractors, nor do we centralize the data using statistics learned in a different set on images.

## E  Pseudocode of CARP in a PyTorch-like Style

```
# NB: number of random prototypes within a block
# K: number of prototypes
```

```python
# NP: number of blocks in the partition, i.e. K // NB
# N: batch size
for x1, x2 in loader:
    # student and teacher branches
    z1, w1 = enc(x1), mom_enc(x1) # [N, K]
    z2, w2 = enc(x2), mom_enc(x2) # [N, K]

    s_logits, t_logits = [z1, z2], [w1, w2]

    # sample cluster indices with no replacement
    rand_proto_ids = multinomial(ones(K), K, False)
    split_proto_ids = stack(split(rand_proto_ids, NB))
    preds_list, targets_list = [], []

    for s_log, t_log in zip(s_logits, t_logits):
        preds = get_logits_gr(s_log, split_proto_ids)
        targets = get_logits_gr(t_log, split_proto_ids)

        preds_list.append(preds)
        targets_list.append(targets)

    loss = loss_fn(preds_list, targets_list)
    # perform gradient descent steps

def loss_fn(s_list, t_list):
    c_loss = consistency_loss(s_list[0], t_list[1]
    c_loss += consistency_loss(s_list[1], t_list[0])

    s = cat(s_list, dim=1)
    t = cat(t_list, dim=1)
    probs = cat([s, t], dim=1).transpose(0, 1)

    e_loss = kl_div(mean(probs, dim=0))
    return c_loss + e_loss

def consistency_loss(s, t):
    loss = einsum("knc,knc->kn", [s, t])
    return -log(loss).mean()

def kl_div(p):
    return mean(log(K) + sum(p * log(p), dim=-1))

def get_logits_gr(logits, proto_ids):
    logits_gr = logits[:, proto_ids.flatten()]
    logits_gr = logits_gr.split(NB, dim=1)
    logits_gr = stack(logits_gr, dim=0)
    return softmax(logits_gr, dim=-1) # [NP, N, NB]
```

