# OpenReview forum: "Representation Learning via Consistent Assignment of Views over Random Partitions"
_NeurIPS.cc/2023/Conference — NeurIPS 2023 poster_

### Official Review · Reviewer_fyYo · 2023-06-27

**Soundness:** 3 good
**Presentation:** 3 good
**Contribution:** 3 good
**Rating:** 5
**Confidence:** 4

**Summary:**

The authors propose a new method for self-supervised learning based on cluster assignments. The method is based on a consistent assignments approach that assigns the same prototype to different views of the same image. To overcome issues of the previous methods that are not well scalable, a divide and conquer approach is proposed. Radom partitions of the learnable prototypes are created each iteration/epoch that allows not only make the training to be stable but also faster. The authors show the efficiency of the approach across a wide range of different datasets and settings.

**Strengths:**

S1: The paper is well written, it’s clearly stated what is novel and what is borrowed from the previous methods, and the explanations of the approach are easy to follow.
S2: The idea of creating subtasks to reduce computation time and avoid collapse sounds novel.
S3: As it is empirically demonstrated, the proposed idea improves the performance over multiple benchmarks on kNN and retrieval tasks while being on-par with linear probe.


**Weaknesses:**

W1: would be interesting to see how the methods work with transformer based architectures. e.g. within moco-v3 framework?
W2: the contributions are not clearly stated apart from the first one: citation [9] in contribution is wrong, it’s SimCLR contrastive approach that should belong to negatives. Moreover, the authors use entropy maximization as a form of normalization which the authors even state as a separate contribution. But this is cannot be a separate contribution because it was used before as a technique to avoid trivial solutions e.g. in [39].
W3: the Fig.7 is not helpful at all. It was not possible to understand what it is depicted there. The text (not caption) is clear though.
W4: In the intro, the authors introduce two classes of SSL. It is too restrictive division and does not cover all SSL methods. Where e.g. MAE lies?
W5: [36] is not really recent while the authors refer to it as “were recently proposed” on L41


**Questions:**

Q1: what if to have hard pseudo-labels instead of soft assignments?
Q2: L218 - it is an interesting observation, could the authors make an analysis and confirmation of that?
Q3: what is the value of \lambda_e, no information in implementation details? how does the performance change if to change this parameter?
Q4: could the authors also add a discussion on the differences to SwAV?


**Limitations:**

The authors address limitations in the form of discussion wrt the previous work based on which they propose the new approach. The section provides nice insights into the limitations of the previous method but also shows rather limited novelty of the current approach over the previous one. The method section is rather describing the previous approach as a new one that can be confusing. However, I find the proposed divide-and-conquer approach very interesting and helpful based on the results. It would be helpful to rewrite partially the paper making more focus on the proposed contribution rather than on the previous method with additional analysis/confirmation of all the statements.

---

> ### Author Rebuttal · Authors · 2023-08-07
>
> > W1: would be interesting to see how the methods work with transformer-based architectures?
>
> We thank the reviewer for the suggestion. It is in our plans to explore how the proposed random partition pretext task would behave with other architectures, such as ViTs. However, we chose ResNets as the main architecture mainly because most SSL methods provide baselines with ResNet backbones. While recent SSL methods are switching to ViT, given our computing budget, running experiments with more than one architecture would be impractical. Thus, we decided to pick one base architecture.
>
> > W2: the contributions are not clearly stated apart from the first one [...] the authors use entropy maximization as a form of normalization [...] But this is cannot be a separate contribution because it was used before as a technique to avoid trivial solutions e.g. in [39].
>
> **Our main contribution is the introduction of the random partition pretext task.** In addition, CARP does not require normalizing the embeddings to a hypersphere (required by other methods), nor does it require mining negatives during training. As the reviewer pointed out, the mean entropy maximization of the probabilities has been used to avoid collapse. However, **CARP does not use it as other methods have**, such as TWIST and PAWS. In CARP, the mean entropy is maximized over the random partitions, i.e., we maximize it over many subproblems constantly changing during training. *We believe that this process removes noisy patterns from the processed data and recovers stable representations due to the stochastic process of the partitions*. The importance of our pretext task to avoid training collapse and improve representations' power is shown in Table B1, appendix.
>
> > W3: the Fig.7 is not helpful at all. It was not possible to understand what it is depicted there. The text (not caption) is clear though.
>
> There is no Fig. 7 in the paper.
>
> If you referred to Fig. 2, it was meant to describe the random partition pretext task. Other clustering-based SSL methods pose the view assignment task over the entire set of learnable prototypes. Conversely, CARP's random partitions break the prototypes into smaller subgroups. Each subset of prototypes is used to optimize a view classification task in parallel. In Fig 2, we have 2 views and 8 prototypes. The prototypes are randomly divided into 2 groups. For each group, (1) we enforce consistent assignment of views and (2) maximize entropy over the batch.
>
> > W4: In the intro, the authors introduce two classes of SSL. It is too restrictive division and does not cover all SSL methods. Where, e.g., MAE lies?
>
> We apologize for overseeing the taxonomy and commit to updating it. Our motivation was to provide a simple and not exhaustive categorization for SSL methods based on joint-embedding architectures. We agree that MAE is an SSL method. However, MAE is an autoencoder (encoder/decoder) model based on pixel reconstructions, which makes it different from joint-embedding methods that operate in the embedding space that we were interested in presenting.
>
> > W5: [36] is not really recent while the authors refer to it as “were recently proposed” on L41
>
> We thank the reviewer for the suggestion. We will reword this phrase in the final version of the manuscript.
>
> > What if to have hard pseudo-labels instead of soft assignments?
>
> CARP’s consistency loss is optimized when the views' probability distributions are identical one-hot vectors (hard assignments). However, sharpening the distributions toward hard assignments is an iterative process and converges in the limit. To address the reviewer's concern, we trained CARP using hard assignments instead of soft ones. We noticed that training becomes unstable and collapses right in the beginning. We hypothesize that the soft assignments have a regularization effect and prevent the embeddings from being assigned to the same prototypes.
>
> > Q2: L218 - it is an interesting observation, could the authors make an analysis and confirmation of that?
>
> In the general rebuttal PDF, we show evidence that the sharpening operation is built-in CARP's loss function. During training, the probability distributions for predictions and targets behave like sharpened one-hot vectors, as shown by the maximum and minimum values of the distribution converging towards one and zero, respectively. The distributions of different sharpening values produce a similar trend.
>
> > what is the value of $\lambda_e$, no information in implementation details? how does the performance change if to change this parameter?
>
> In practice, the value of $\lambda$ is 1. One of the benefits of our proposed random partition pretext task is a regularization effect that helps prevent collapsed solutions. CARP does not require tuning the $\lambda$ parameter that defines the contribution of the entropy term to the final loss.
>
> We thank the reviewer for pointing out this missing piece. *We will add a description of this parameter in the final version of the manuscript.*
>
> > Could the authors also add a discussion on the differences to SwAV?
>
> The most important difference between CARP and SwAV is the way they avoid trivial solutions. While SwAV uses a non-differential iterative module (Sinkhorn–Knopp) to balance the target predictions over the learned prototypes, CARP uses the random partition pretext task combined with the mean entropy maximization. Plus, SwAV requires normalized embeddings to avoid NaNs during training; CARP does not. SwAV enforces consistent assignment between views using the cross-entropy loss and requires extra hyperparameters to sharpen the probability distributions. CARP, on the other hand, **has the sharpening of the probabilities built into its consistency loss**. CARP's consistency loss is optimized when the probability distributions of the views are identical one-hot vectors. CARP's consistency loss avoids extra hyperparameters and sharpens the distributions by design.

---

> > ### Comment · Reviewer_fyYo · 2023-08-19
> >
> > Thank you for the authors' detailed response. After reading the rebuttal and the comments of the other reviewers, my concerns have been  addressed and thus I maintain towards the acceptance of this work.

---

> > > ### Author Response · Authors · 2023-08-21
> > > **Thanks to reviewer's work**
> > >
> > > We thank the reviewer for the relevant work reviewing our manuscript and for the final position towards acceptance of our work.

---

### Official Review · Reviewer_osfH · 2023-07-03

**Soundness:** 2 fair
**Presentation:** 3 good
**Contribution:** 2 fair
**Rating:** 3
**Confidence:** 5

**Summary:**

This work extends the existing work about consistent assignment for representation learning by introducing random partitions. Specifically, when the number of prototypes is large in clustering-based self-supervised learning, the work shows that CARL [36] cannot handle the loss and the regularization well. To mitigate the problem, CARP is proposed to obtain subsets of prototypes and then apply CARL on the partitions of prototypes. Standard evaluation protocol is included to compare the proposed method to benchmark methods.

**Strengths:**

1. The clustering-based method shows the potential for self-supervised learning.
2. The strategy of random partition can avoid the problem from a large number of prototypes.
3. Experiments are extensive on diverse data sets.

**Weaknesses:**

1. As a closely-rated work, the comparison to CARL lacks, which makes the contribution of this work unconvincing. Moreover, the original paper of CARL shows that it works well with 10000 prototypes, which contradicts the claim of this work.
2. The performance of clustering-based method in the comparison is out of date. CoKe [1] as an online clustering method demonstrates 74.9% and 76.4% linear probing performance on ImageNet with two-view and multi-view augmentations, respectively. Moreover, CoKe does not request an additional momentum encoder and a sharpening temperature for the pseudo-label from the other view, which makes the contribution of this work less significant.
3. While CARP shows a competitive performance with less training epochs, it is better to show if it can achieve a better performance with the long training strategy.

[1] Unsupervised Visual Representation Learning by Online Constrained K-Means. CVPR 2022.

**Questions:**

Besides the weakness, my major concern is about the limited contribution compared to CARL.

**Limitations:**

Yes.

---

> ### Author Rebuttal · Authors · 2023-08-07
>
> > [...] my major concern is about the limited contribution compared to CARL.
>
> The main difference between CARL and CARP is the random partition pretext task and its positive effects on training SSL models. As described in Section 3.1, training a system like CARL is difficult due to stabilities, requiring tuning the entropy term and dealing with the negative trade-off. **Our pretext task diminishes the trade-off effect between the two losses and allows stable training with better outcomes.** Moreover, the entropy regularizer in CARP is also optimized over the prototypes within the subsets (which differs from CARL), contributing to stable training and high-quality representations (high scores in transfer learning benchmarks). Another part of our contribution is the **understanding of how we can stabilize training and avoid collapsed solutions through a stochastic sampling process** that is achieved with the proposed random partition pretext task.
>
> > [...] the comparison to CARL lacks, which makes the contribution of this work unconvincing. [...] CARL [...] works well with 10000 prototypes, which contradicts the claim of this work.
>
> Even though CARL can be trained with many prototypes, **one needs to carefully tune the weight contribution for the entropy term to avoid training collapse.** As discussed in Section 3.1, **representation quality decreases if the entropy term is too high, and collapse might happen if it is too low.**
>
> We show evidence of this effect in Table B.2 in the appendix. As we increase the size of the partitions, performance decreases. The last column (Table B.2) is equivalent to **not having random partitions** (i.e., CARL), where we see collapse due to a large number of prototypes. The random partition pretext task addresses these limitations by improving overall performance and increasing training stability.
>
> CARL is focused on small-scale datasets such as Cifar-10/100 and STL-10, while our work's primary pre-training data is ImageNet. Nevertheless, to address the reviewers' concerns, we took CARL's official code (gitlab.com/mipl/carl), and pre-trained it on ImageNet for 200 and 400 epochs using 3000 prototypes (we experienced collapse with 10000). The results below show that CARP outperforms CARL on linear evaluation by considerable margins.
>
> \begin{array} {rr}\hline
> Method&Ep&Acc&kNN\\\\
> \hline
> CARL&200&65.3&59.4\\\\
> &400&70.4&62.8\\\\
> CARP&200&74.0&66.5\\\\
> &400&75.0&67.7\\\\
> \hline
> \end{array}
>
> > The performance of clustering-based method in the comparison is out of date. CoKe [1] as an online clustering method demonstrates 74.9% and 76.4% linear probing performance [...]
>
> To address the reviewer's concern, we took the officially released 800 epoch model from CoKe's repository and ran our benchmark evaluations. We present the results below.
>
> k-NN transfer learning (k=20), paper's Table 1
>
> \begin{array}{lrcrrrrrrrrrrr}
> \hline
> Method&Ep&Pets&Flowers&Aircraft&Cars&Country&Food&STL&GTSRB&Avg @k&&&\\\\
> &&&&&&&&&&10&20&100&200\\\\
> \hline
> CoKe&800&79.5&79.5&27&22.4&14.6&58.9&95.7&60.4&57.1&57.4&57.1&56.3\\\\
> CARP&400&86.8&80&42.1&33.5&12.3&58.4&95.9&75.3&60.4&60.5&59.7&59.2\\\\
> \hline
> \end{array}
>
> Clustering evaluation, paper's Table 2
>
> \begin{array}{lrrrrrrrrrrrr}
> \hline
> Method&ImgNet&&&Cifar10&&&Cifar100&&&GTSRB&&\\\\
> &NMI&AMI&ARI&NMI&AMI&ARI&NMI&AMI&ARI&NMI&AMI&ARI\\\\
> \hline
> CoKe&68.9&45.6&21.3&45.9&45.8&34.2&51.9&45.2&19.5&49.4&47.1&13.7\\\\
> CARP&70.3&48&23.9&49&48.9&38.7&54.5&48.2&23.1&54.8&52.7&19.6\\\\
> \hline
> \end{array}
>
> CoKe performs well when trained and evaluated on ImageNet. However, on transfer tasks, **CARP outperforms CoKe by large margins on both k-NN and k-means evaluations.**
>
> > Moreover, CoKe does not request an additional momentum encoder and a sharpening temperature for the pseudo-label from the other view, which makes the contribution of this work less significant.
>
> We'd like to emphasize that CoKe does sharpen the probability distributions in its loss function. If we look closer at CoKe's code (github.com/idstcv/CoKe/blob/main/main_multi_view.py#L351C33-L351C33), we can see two losses (1) loss_pred and (2) loss_proj. The first is a hard loss, where predictions are sharpened. The second is a soft loss, where predictions and target distributions are sharpened before softmax (refer to parameter $coke_t$).
>
> In fact, CARP and CoKe's loss functions differ significantly. While CoKe uses a sharpened cross-entropy loss for both hard and soft losses, **CARP proposes a consistency loss where the goal is to minimize the negative log of the product of the views probability distributions**. In the general rebuttal PDF, we show evidence that the sharpening operation is built-in CARP's loss. During training, the probability distributions for predictions and targets are automatically sharpened toward one-hot vectors.
>
> Also, CARP's contributions **go beyond architectural and implementation details such as momentum encoders and use of temperatures** and mainly **focus on a novel approach to learning representations through prototypes that stabilizes training, avoids collapses, and yields transferable representations.**
>
> > While CARP shows a competitive performance with less training epochs, it is better to show if it can achieve a better performance with the long training strategy.
>
> To address the limitations pointed out by the reviewer, we trained CARP for 800 epochs with multi-crop. CARP archives 75.9% linear probing on ImageNet, which beats methods such as DINO and SwAV (75.3%). While we understand the reviewer's point of view, we emphasize that *an efficient method is cheaper, allows faster training with reduced hardware costs and emissions (democratizing research), and has strong inductive biases* (allowing the model to extract patterns quickly).

---

> > ### Comment · Reviewer_osfH · 2023-08-14
> >
> > I appreciate the efforts of the rebuttal. However, it did not address my major concerns.
> > 1. For the comparison with CARL, the experiment on ImageNet has collapsing as stated in rebuttal. However, it can be avoided by tuning the weight of the entropy. Since CARL does not have the results on ImageNet, is the parameter of CARL tuned for ImageNet? Moreover, Table 1 in the original paper of CARL shows that it works well with 10,000 prototypes on CIFAR-100. Therefore, 10,000 prototypes should not be a big issue on ImageNet. Tuning the parameter for CARL on ImageNet can be expensive, so it is better to have CARP on CIFAR-100 for a fair comparison. Moreover, the proposed method eliminates the parameter for the entropy while introducing an additional parameter for the partition size, which may not save tuning efforts.
> > 2. For the comparison with CoKe, I note that the CoKe with multi-crop augmentation is adopted for the comparison while CARP has a different setting. Compared with CARP with multi-crop augmentation as in Table 1 in the submission, the gap is quite marginal, i.e., 0.3% on Avg@k. A fair comparison is essential to draw any conclusion. Besides, the phenomenon for the degenerated performance from the multi-crop augmentation is interesting and worth an additional discussion.
> >
> > Some minor issues.
> > 1. Many deep clustering methods also report clustering results on the benchmark data sets as in Table 2. For example, CoKe reports 76.6% NMI on CIFAR-10 as a clustering method, which is much better than the result in this work. The setting of clustering in this work can be highlighted in the caption of Table 2 to avoid misunderstanding.
> > 2. Unlike [7,8] that have different temperatures for pseudo labels and predictions, CoKe has the same temperature and the label is not sharpened. While this work does not have the temperature, the statement can be more accurate.

---

> > > ### Author Response · Authors · 2023-08-15
> > > **Reply to rebuttal comments 1/2**
> > >
> > > > Since CARL does not have the results on ImageNet, is the parameter of CARL tuned for ImageNet.
> > >
> > > As the reviewer mentioned, CARL was not trained on ImageNet. We did the hyperparameter search for the number of prototypes due to our experience executing CARL. When CARL was trained with 10000 prototypes, training collapsed. We kept the entropy weight as 2 (following CARL’s guidelines for other datasets) and reduced the number of prototypes to 3000 (where we observed no collapse).
> > >
> > > To fully address the reviewer’s concerns, we pre-trained multiple setups of CARL on ImageNet for 100 epochs. We fixed the number of prototypes as 10000 and ablated different values of lambda (entropy weight). Results are below.
> > > \begin{array} {rrrrrr}
> > > \hline \lambda &1&2&3&4&5\\\\
> > > \hline
> > > CARL&C&C&65.1&64.8&63.9\\\\
> > > \hline
> > > \end{array}
> > > As discussed, using 10000 prototypes causes collapse (**C**), and the workaround is to increase the entropy weight. For reference, CARP 100 epoch model with 2 views achieves 69.7%.
> > > As discussed in Section 3.1, learning that many prototypes at once with the entropy term, as in CARL, leads to underperformance.
> > >
> > > > Table 1 [...] CARL shows that it works well with 10,000 prototypes on CIFAR-100. Therefore, 10,000 prototypes should not be a big issue on ImageNet.
> > >
> > > We respectfully disagree with the reviewer on this point. The training of CARL on a much simpler dataset with only 100 classes cannot be extrapolated to a more complex dataset with 1000 classes (10 times more). Moreover, CARL’s paper discusses the relation of the number of prototypes and the number of classes (Fig. 3 in CARL’s paper) which is in line with the observed behavior in our experiments. **In CARP, we show that there is a lack of generalization in the usage of prototypes and propose a solution.**
> > >
> > > > [...] it is better to have CARP on CIFAR-100 for a fair comparison.
> > >
> > > The results comparing CARP and CARL on CIFAR-100 are below.
> > > \begin{array}{rrrrrrr}
> > > \hline
> > > & Cifar10&&Cifar100&&STL10&\\\\
> > > \hline
> > > Ep. &100&200&100&200&100&200\\\\
> > > \hline
> > > CARL&73.39&78.94&42.91&48.85&76.9&81.95\\\\
> > > CARP&74.84&79.52&44.67&50.64&78.05&82.44\\\\
> > > \hline
> > > \end{array}
> > >
> > > We pre-trained CARP on CIFAR-10/100 and STL-10 following the CARL’s guidelines (Table 2 on CARL’s paper). We report average top-1 (linear probing) results across 3 independent runs (same as CARL). The number of prototypes was set to 100, 300, and 300 for Cifar-10,-100, and STL-10, respectively (same as CARL), and the partition size was set to 50 for all datasets (no tuning was done to select this partition size). We report models trained for 100 and 200 epochs. CARP outperforms CARL in all datasets. These experiments showed that CARP works well for simpler datasets as well as in ImageNet. By default, CARP uses $\lambda=1$ while CARL uses larger values to avoid collapse, e.g., $\lambda$=2, which explains CARP’s improvements.
> > >
> > > > Moreover, the proposed method eliminates the parameter for the entropy while introducing an additional parameter for the partition size, which may not save tuning efforts.
> > >
> > > We agree with the reviewer that our method introduces a new hyperparameter (the partition size) and avoids the tuning of the entropy weight. However, **the entropy weight and the partition size have different tuning difficulties**. As discussed in Section 3.1, without the random partition pretext task, training collapses if the entropy term is too small, and accuracy is suboptimal if the entropy term is too high. On the other hand, CARP is robust to the choice of partition size, as shown in Table B.2 (appendix).
> > >
> > > > [...] Compared with CARP with multi-crop augmentation as in Table 1 in the submission, the gap [with CoKe] is quite marginal, i.e., 0.3% on Avg@k. A fair comparison is essential to draw any conclusion.
> > >
> > > As pointed out by the reviewer, when comparing CoKe with CARP with multi-crop, CARP still outperforms CoKe by a small margin (even though CoKe was trained for 800 epochs and CARP for 400). To clear the reviewer's concerns, we extended the previous comparison against CoKe to include all instances of CARP and an additional instance of CoKe trained for 800 epochs w/o multi-crop. CARP consistently outperforms CoKe by large margins.
> > >
> > > \begin{array}{lrcrrrrrrrrrrr}
> > > \hline
> > > Method&Ep&Pets&Flowers&Aircraft&Cars&Country&Food&STL&GTSRB&Avg @k&&&\\\\
> > > &&&&&&&&&&10&20&100&200\\\\
> > > \hline
> > > CoKe &1000&81.3&75.3&29.3&22.6&13.3&60&95.7&64.2&55&55.2&54.1&53.4 \\\\
> > > CoKe (mc) &800&79.5&79.5&27&22.4&14.6&58.9&95.7&60.4&57.1&57.4&57.1&56.3\\\\
> > > CARP&200&86.8&78.2&38.9&29.8&12.2&58.4&95.5&73.7&59.2&59.2&58.5&57.9\\\\
> > > &400&86.8&80&\textbf{42.1}&\textbf{33.5}&12.3&58.4&95.9&75.3&60.4&60.5&59.7&59.2\\\\
> > > &800&\textbf{87.3}&\textbf{81.2}&41.1&33.2&13.6&61.2&\textbf{97}&\textbf{76.4}&\textbf{61.2}&\textbf{61.4}&\textbf{60.4}&\textbf{59.7}\\\\
> > > CARP (mc)&200&78.7&79.7&35&26.6&\textbf{14.5}&61.8&95.5&64.7&57.1&57.1&55.9&55\\\\
> > > &400&83.9&80.3&34.8&27.1&14.2&\textbf{62.9}&95.5&62.8&57.6&57.7&56.8&56\\\\
> > > \hline
> > > \end{array}

---

> > > ### Author Response · Authors · 2023-08-15
> > > **Reply to rebuttal comments 2/2**
> > >
> > > > [...] the phenomenon for the degenerated performance from the multi-crop augmentation is interesting and worth an additional discussion.
> > >
> > > We agree with this observation. We expect to study it further in the future. As it is now, this understanding is out of the scope of this work.
> > >
> > > > [...] For example, CoKe reports 76.6% NMI on CIFAR-10 as a clustering method, which is much better than the result in this work. [...]
> > >
> > > We highlight that CoKe’s results on CIFAR-10 were reported with an inter-dataset setup, meaning that it was trained and evaluated on CIFAR-10. In our experiments, CARP was pre-trained on the ImageNet dataset and evaluated on CIFAR-10 (transfer learning). The latter is a more challenging setup that evaluates the generalization and robustness of the learned representations. This explains the difference observed by the reviewer.
> > >
> > >
> > > > [...] The setting of clustering in this work can be highlighted in the caption of Table 2 to avoid misunderstanding.
> > >
> > > We reported the evaluation protocol in Appendix D.4 (as mentioned in Section 5.2). In addition, we will improve the caption in Table 2 to better convey the evaluation protocol..
> > >
> > > > Unlike [7,8] that have different temperatures for pseudo labels and predictions, CoKe has the same temperature and the label is not sharpened. [...]
> > >
> > > We do not understand the reviewer’s comment given that CoKe clearly uses the temperature. First, all sets of variables are sharpened here:  github.com/idstcv/CoKe/blob/main/coke/builder_double_view.py#L98C1-L102C76. Then, used the in the softmax here: github.com/idstcv/CoKe/blob/main/main_double_view.py#L321-L323. Then, in the loss function here: github.com/idstcv/CoKe/blob/main/main_double_view.py#L330C1-L337C97.
> > >
> > > We stated the differences between CARP’s and CoKe’s loss functions and the comparison against other methods, like SwAV [7] and DINO [8]. We would appreciate it if the reviewer could provide additional information about the concern raised in this question.
> > >
> > > > [...] While this work does not have the temperature, the statement can be more accurate.
> > >
> > > We will review our manuscript to improve the accuracy of how the new pretext task produces the sharpening and its relation with the temperature. As discussed in the other reviews, we will include the empirical evidence that the random partition process sharpens the prediction by design, as shown in the PDF rebuttal document.
> > >
> > >
> > > We hope to have addressed all the open questions about our work.

---

> > > > ### Comment · Reviewer_osfH · 2023-08-15
> > > >
> > > > Thanks for the additional experiments for the clarify. The baseline CARL seems work well on ImageNet. Considering that CARL does not have the additional momentum encoder while that helps a lot for the proposed CARP, is the gain from the architecture rather than the algorithm?

---

> > > > > ### Author Response · Authors · 2023-08-16
> > > > > **Reply about further considerations about CARL**
> > > > >
> > > > > > Considering that CARL does not have the additional momentum encoder while that helps a lot for the proposed CARP, is the gain from the architecture rather than the algorithm?
> > > > >
> > > > > Indeed, **CARL works** on ImageNet **once the entropy term is properly tuned to avoid collapse**. Regarding the momentum encoder, we have ablated the importance of the momentum encoder in the CARP architecture; please refer to Table B.2 in the appendix.
> > > > >
> > > > > While the momentum encoder increases the final performance, the gains are not that significant. Indeed, CARP trained for 100 epochs (no multi-crop) **without** the momentum encoder performs at 68.6% (1.1% decrease in ImageNet linear probing), which suggests that **the extra performance boost, compared to CARL, comes from the random partition pretext task**.
> > > > >
> > > > > Indeed, the practical results suggest there is a trade-off between the entropy term and the consistency loss. Thus, **the higher entropy term that prevents CARL from collapsing also contributes to the suboptimal performance**. **Our proposed pretext task alleviates this trade-off and allows CARP to perform better**. This trade-off is discussed in Section 3.1 (our paper) and also evidenced in CARL's paper Section 4.1.

---

> > > > > > ### Author Response · Authors · 2023-08-17
> > > > > > **CARL with a momentum encoder**
> > > > > >
> > > > > > To fully address the reviewer's concern, we trained CARL with a momentum encoder for 100 epochs, with 10000 prototypes and a $\lambda=3$ (same configuration as the table above). **CARL with a momentum encoder scored 65.6% linear probing on ImageNet, which represents an improvement of 0.5% over the instance (w/o momentum encoder) reported in the table above**. The momentum encoder was updated using the same hyperparameters used in CARP. The momentum encoder did not contribute to further improvements. The result suggests that the performance gap between CARP (69.7%) and CARL (65.6%) comes from our random partition pretext task and the smaller weight in the entropy term used in CARP.

---

> > > > > > > ### Author Response · Authors · 2023-08-20
> > > > > > > **Ending of discussion period**
> > > > > > >
> > > > > > > We thank the reviewer for the productive discussions and for helping us improve our work. Since the discussion period is ending, we would like to ask the reviewer to conclude the assessment of our work. Moreover, if the reviewer feels that the main concerns (especially the comparisons to CoKe and CARL) have been addressed, we ask the reviewer to update the score to reflect such a position.

---

### Official Review · Reviewer_FHva · 2023-07-06

**Soundness:** 3 good
**Presentation:** 3 good
**Contribution:** 2 fair
**Rating:** 6
**Confidence:** 4

**Summary:**

This paper works on self-supervised representation learning. Under the setting of consistent clustering assignment between augmented views (SwAV-like), the authors found that when the number of prototypes is significantly larger than the batch size, the commonly used technique for avoiding trivial solutions fails. And they thus propose to randomly partition the prototypes into multiple subgroups to avoid the trivial solution. Experiments on extensive benchmarks, especially retrieval-based ones, validate the effectiveness of the proposed method.

**Strengths:**

*Originality:* The major insight of this paper: entropy regularization fails when the prototype number is significantly greater than the batch size ($K \gg N$) is novel, and the solution that divides the prototypes into multiple subgroups, is straightforward and verified to be useful. Other components of this framework resemble multiple previous works (*e.g.*, SwAV, DINO, MSN).

*Quality:* The idea is clearly formulated and presented, and the method is well evaluated in extensive experiments, making it in shape of a solid paper.

*Clarity:* The idea is straightforward and the delivery is clear enough, reading it is smooth and I do not find trouble in understanding.

*Significance:* It tackles the assignment strategy in clustering-based self-supervised learning, and provides a solution for $K \gg N$, which is somehow useful.

*Reproducibility:* Code is provided in the supplementary material to facilitate reproduction.

**Weaknesses:**

*Motivation*: Why do we need so many prototypes during pre-training? The major stream of self-supervised learning is to enlarge the batch size for better representation learning. The prototype number, in contrast, has not shown the necessity for very large numbers yet. In fact, in DINOv2 they have reduced $K$ from 65536 in DINOv1 to 4096, which is competent for learning good representations. Moreover, large number of prototypes means high computational cost in pre-training. And after pre-training, they are simply dropped and not used in downstream tasks. So why do we need so many prototypes?

*Technical Contribution:* In terms of technical contribution, this work proposes to randomly divide the prototypes into subgroups. The idea is clear and straightforward but the contribution might be limited.

*Evaluation*: The evaluation is mainly focused on $k$-NN or linear probing on retrieval-based benchmarks. However, for one thing, a good deep representation does not have to be linear, and performance under full fine-tuning (transfer learning) might be of higher interest. For another, one might be more interested in generic classification benchmarks (ImageNet) and dense prediction downstream tasks.

**Questions:**

- I noticed that different tables may refer to different variants of CARP (or CARP w/ mc). Why is it and why multiple cropping does not help in e.g., Tab. A.1?
- One advantage of clustering-based methods is that they tend to suit ViTs better (e.g., in DINO). Why is this work restricted in ResNets?

**Limitations:**

limitations are discussed in Sec. 7

---

> ### Author Rebuttal · Authors · 2023-08-07
>
> > Why do we need so many prototypes during pre-training? [...]
>
> Based on our practical experience, **the optimal number of prototypes highly depends on the number of hidden classes of the dataset.**  Due to ImageNet's high number of classes (1000), in practice, a higher number of prototypes produces better outcomes.
>
> We ablated the number of prototypes in Section B.1 in the appendix. Practical results suggest that over-clustering is beneficial. **However, CARP proved to be robust to the number of prototypes.** Indeed, the **difference in k-NN performance when learning 65536 prototypes as opposed to 4096 prototypes is only 0.5%, which agrees with the reviewer's point about Dino-v2**.
>
> While we agree with the reviewer's concerns about the large number of prototypes, our empirical evidence suggests that there is a trade-off that must be considered.  However, to provide conclusive evidence and best practices, more work understanding this trade-off is needed.  However, this is out of the scope of our current proposal.
>
> > In terms of technical contribution, this work proposes to randomly divide the prototypes into subgroups. The idea is clear and straightforward but the contribution might be limited.
>
> In summary, our main contributions are the proposal of a stochastic partitioning process to improve the learned representations' performance on downstream tasks, as demonstrated in our comprehensive evaluation protocol.  Moreover, we empirically show that this process reduces the noise on the learned representations, stabilizes the training, and avoids its collapse.
>
> > The evaluation is mainly focused on k-NN or linear probing on retrieval-based benchmarks. However, for one thing, a good deep representation does not have to be linear, and performance under full fine-tuning (transfer learning) might be of higher interest. For another, one might be more interested in generic classification benchmarks (ImageNet) and dense prediction downstream tasks.
>
> We strived to design an evaluation protocol focused on two main aspects.  First, we wanted to evaluate the learned representation in transfer learning scenarios.  Second, the evaluation protocol should be diverse, including many distinct datasets with varying difficulty levels.
> Our evaluation protocol includes over 15 datasets across many standard protocols, such as linear evaluation, few-shot classification, k-NN, k-means, image retrieval, and copy detection.
>
> We agree with the reviewer that a good representation does not need to be linear. Nevertheless, **linear evaluations test the representations power** by using a simpler classifier, **giving a better intuitive sense of how good the representation is since it does not rely on other components**, such as the backbone architecture or the myriad of hyperparameters used to fine-tune.  A good example is linear probing.  Even though it is a linear protocol, many hyperparameters can influence the final result.   Consequently, comparing linear performance between methods that used different training strategies becomes unreliable.
>
> A good example is NNCLR [1], where the logistic regressor on top of the frozen representation is trained with the LARS optimizer and large batch sizes.  Based on our experiments, this configuration produces significantly better final accuracy than the approach proposed by MoCo (vanilla SGD with small batch sizes).  **With Linear evaluations based on k-NN** of k-Means, **the effect of external hyperparameters, choices of optimizers, or even the batch size, is reduced**, which in our view, **makes the comparison fairer.**  Nevertheless, we support evaluation protocols that test different aspects of the representation.
>
> [1] Dwibedi, Debidatta, et al. "With a little help from my friends: Nearest-neighbor contrastive learning of visual representations." ICCV. 2021.
>
> > I noticed that different tables may refer to different variants of CARP (or CARP w/ mc).
> > Why is it[…]
>
> For our evaluation, we trained 4 instances of CARP in total.  CARP 200 and 400 epochs without multi-crop augmentation and CARP 200 and 400 epochs with multi-crop (w/ mc).  In the tables in the main text, we report the instances of CARP that performed best in the downstream tasks.  Indeed, we did the same for the competing methods.  In total, we compared CARP against 11 existing SSL methods.  However, in the main text, we reported only a subset that performed well on a given downstream task due to space constraints.  **We showed the full evaluation for all instances of CARP and all competing SSL methods in the appendix.**
>
> >  and why multiple cropping does not help in e.g., Tab. A.1?
>
> As for the performance using multi-crop, it also caught our attention.  In short, **we suspect that multi-crop augmentation may be causing many SSL methods to overfit to ImageNet's training data distribution.**  As a result, we may see higher performance scores for linear probing on the same ImageNet, but modest transfer learning performance to other datasets and tasks.  However, further experiments are required to fully understand the impact of multi-crop and the decrease in performance seen in our experiments.  These evaluations are out of the scope of our current proposal, though.
>
> > One advantage of clustering-based methods is that they tend to suit ViTs better (e.g., in DINO). Why is this work restricted in ResNets?
>
> It is in our plans to explore how the proposed random partition pretext task would behave with other architectures, such as ViTs.  However, we chose ResNets as the main architecture mainly because most SSL methods provide baselines with ResNet backbones.  Indeed, recent SSL methods are switching to ViTs. However, given our computing budget, running so many long experiments with more than one architecture would be impractical.  For this reason, we decided to pick one base architecture and do the best we could regarding training, ablations, and evaluations to make our proposal clear and robust.

---

> > ### Comment · Reviewer_FHva · 2023-08-15
> >
> > Thanks to the authors for their efforts in providing the response. My major concern is still about the motivation: *when and why do we need so many prototypes?* The authors claim that a higher number of prototypes produces better outcomes on datasets that have more hidden classes. Empirically this is reasonable for datasets that are as big or smaller than ImageNet ILSVRC12, but for them, the techniques proposed in this work are also not necessary as agreed by the authors. Hope to see the motivation validated in larger-scale datasets.

---

> > > ### Author Response · Authors · 2023-08-16
> > > **Reply to rebuttal comments**
> > >
> > > > [...] about the motivation: when and why do we need so many prototypes?
> > >
> > > **(Why)** Our requirement for more prototypes stems from their ability to leverage local information derived from our learning objective (unsupervised local view agreement). This local data is insufficient to grasp coarser (global) clusters, but it effectively captures changes within the data manifold, as evidenced by our findings.
> > >
> > > Based on our empirical experiments and intuitive understanding of our method, here's a breakdown of our earlier point:
> > >
> > > With fewer prototypes, each one must encompass a larger set of images exhibiting substantial variations. Consequently, these prototypes must encapsulate more abstract concepts to maintain coherence among the represented images. Conversely, when the number of prototypes matches the training set's image count, the clusters become specific representations, resembling a k Nearest Neighbor problem. Our experiments revolve around investigating how the learned representations behave as we alter the prototype numbers (while keeping them below the sample count).
> > >
> > > Due to the localized nature of our learning objective, operating on local views of the same image, learning more abstract representations becomes more challenging. To alleviate this challenge, we increase the prototype count. This adjustment lessens the demand for prototypes to grasp intricate abstractions, allowing them to focus on specific characteristics sufficient to represent the smaller set of images with minor variations.
> > >
> > > This over-clustering of the data space effectively harnesses our sole source of information: local information. The large number of prototypes, coupled with a stochastic random partitioning process, induces a noise-filtering mechanism that enhances representation robustness, as our experiments illustrate.
> > >
> > > **(When)** We aim to overparameterize within the bounds of having more prototypes than or equal to the true class count and fewer prototypes than the data points in the training set.
> > >
> > > In the unsupervised scenario, the class count remains unknown. Thus, our task involves exploring the dataset to strike the right balance among prototype count, the unknown class quantity, and the sample number. This process resembles k-means clustering, where data exploration guides the identification of optimal cluster patterns.
> > >
> > > Setting up fewer prototypes than true classes results in prototypes acting as higher-level categories. Conversely, matching the prototype and class counts yields generalized representations that remain consistent despite class variations. Exceeding true class numbers leads to more specific representations capturing local changes within the data manifold.
> > >
> > > > [...] for [datasets as big as ImageNet] the techniques proposed in this work are also not necessary as agreed by the authors
> > >
> > > We highlight that the **main point is not about the number of prototypes to be as high as possible**, but rather to **have a number high enough that allows us to exploit the locality of the views** (through the learning objective) **while supporting the number of classes** (and their complexity).
> > >
> > > We clarify that **we don’t mean** (in our rebuttal) **that the high number of prototypes is not necessary to obtain good performance** (as mentioned by the reviewer in this comment). But rather, we mean that **there is a trade-off** between the number of prototypes and classes (and their complexity through their multi-modality) **that must be understood** and considered to select the optimal number of prototypes that will ideally represent each class. However, in our unsupervised setup (and in real-world scenarios where the data is complex and hard to define), this number is not so easy to define and requires experimental evidence to set.
> > >
> > > In addition, we show in our experiments (Table B.1 appendix) that CARP is robust to the number of prototypes, and the difference in performance from 65k to 4k prototypes is marginal, which shows CARP's capacity to learn more abstract or more specific prototype representations. We hypothesize that this minimal change is related to the information we get from our learning objective. However, as we decrease the number of prototypes even more (< 4k), the learning problem becomes more complex because the prototypes require a higher level of abstraction, and our self-supervised objective does not provide a sufficient level of information. Consequently, the performance decreases. The understanding of this trade-off and to automatically find ways of keeping the relevant prototypes is future work that we intend to tackle. However, it is out of the scope of our current proposal. Nevertheless, we agree with the reviewer about its relevance.
> > >
> > > > Hope to see the motivation validated in larger-scale datasets.
> > >
> > > We intend to evaluate our method on larger datasets for our future work. However, to provide such results during the rebuttal is impossible, given our resources.

---

> > > > ### Comment · Reviewer_FHva · 2023-08-17
> > > >
> > > > Thanks for the authors' reply and hope to see this work validated at scale, I have decided to upgrade my score.

---

> > > > > ### Author Response · Authors · 2023-08-21
> > > > > **Thanks to reviewer's work**
> > > > >
> > > > > We thank the reviewer for the valuable questions and suggestions about our work and for increasing the final score once given sufficient evidence.

---

> > > ### Author Response · Authors · 2023-08-16
> > > **About DINO v2 number of prototypes.**
> > >
> > > Based on the following evidence from Dino v2 repository, we would like to emphasize that DINO v2 uses **65536 prototypes for ImageNet-1K pre-training** (github.com/facebookresearch/dinov2/blob/main/dinov2/configs/ssl_default_config.yaml#L45), not 4096 as previously stated. Moreover, to pre-train on even larger scale datasets such as ImageNet 22k (github.com/facebookresearch/dinov2/blob/main/dinov2/configs/train/vitg14.yaml#L2), **DINO v2 uses 131072 prototypes**. These numbers are aligned with our experimental results and intuitive explanation in the comment above.

---

### Official Review · Reviewer_VZFy · 2023-07-07

**Soundness:** 4 excellent
**Presentation:** 4 excellent
**Contribution:** 3 good
**Rating:** 7
**Confidence:** 4

**Summary:**

This paper addresses a collapsing problem that arises in clustering-based contrastive learning. To resolve the problem, the paper proposes an improved version of consistent assignment in CARL, utilizing a strategy of random partitioning. In particular, the original consistent assignment loss exhibits a stability issue when a large number of prototypes are employed; thus, it is challenging to determine proper hyperparmeters to ensure both stabilization and strong performance. This paper introduces a method of random partitioning among the trainable prototypes and applies the consistent assignment loss across each partition. Such approach effectively mitigates the instabilty problem associated with a large number of clusters while maintaining the benfits from using numerous clusters. Importantly, the proposed method does not introduce any additional hyper-parameters, such as trade-off parameter or sharpening temperature. The experimental results demonstrate the effectiveness of the proposed method in various downstream tasks such as transfer learning, clustering, image retrieval, copy-detection, few-shot classification, and linear/k-NN evaluation.

**Strengths:**

* The method seems simple yet effective, achieving robust performance and preventing collapse without the need for additional hyperparameters.
* The experiments are conducted across various downstream tasks, including transfer learning, clustering, image retrieval, copy-detection, few-shot classification, and linear/k-NN evaluation; the method consistently demonstrates strong performance.
* The method has been meticulously ablated (in main text and supplementary) with various implementation choices thoroughly examined.
* The paper is well-composed and easy to follow.

**Weaknesses:**

* The discussion and comparison to other related baseline methods [1, 2, 3], which similarly employ the entropy of the mean probabilities of a batch while addressing a stabilization of it, seem to be missing.

[1] Self-Supervised Learning by Estimating Twin Class Distributions, arxiv 2021 \
[2] Unsupervised Visual Representation Learning via Mutual Information Regularized Assignment, NeurIPS 2022 \
[3] Masked Siamese Networks for Label-Efficient Learning, ECCV 2022

**Questions:**

### Suggestion
* Please address the concerns listed in the weaknesses.
* Given that all experiments are conducted using ResNet-50, experimental results with Vision Transformers (ViTs) shall strengthen the contributions of the paper.


### Question
* Why is it important or beneficial to avoid using non-differentiable modules, such as Sinkhorn-Knopp, for generating target cluster assignments? (L4-5, L99-100, L101-103)

**Limitations:**

The paper describes a limitation of the proposed method that the representations learned through CARP do not transfer well to dense prediction tasks. This limitation seems logical, as the proposed method simulates smaller pseudo-classification problems, which might be more suitable for downstream tasks such as classification, clustering, and retrieval.

---

> ### Author Rebuttal · Authors · 2023-08-07
>
> > The discussion and comparison to other related baseline methods [1, 2, 3], [...], seem to be missing.
>
> To address the concerns regarding a proper comparison to the other suggested methods, we took the pre-trained models from their respective official repositories and ran the same benchmark used in the paper for all methods. *These results support the conclusion that CARP learns representations that generalize to future tasks.* We detail the results below.
>
> We took the 400 epoch multi-crop MIRA (github.com/movinghoon/MIRA) and the 850 epoch multicrop TWIST (github.com/bytedance/TWIST/tree/main), both with ResNet backbones. We did not benchmark MSN (github.com/facebookresearch/msn) because it only provides ViTs backbones which invalidates a fairer comparison with CNN-based encoders. We display the results in the table below.
>
> Clustering evaluation, paper's Table 2
>
> \begin{array}{lrrrrrrrrrrrr}
> \hline
> Method&ImgNet&&&Cifar10&&&Cifar100&&&GTSRB&&\\\\
> & NMI&AMI&ARI&NMI&AMI&ARI&NMI&AMI&ARI&NMI&AMI&ARI\\\\
> \hline
> TWIST&\textbf{72.7}&\textbf{52.4}&\textbf{28.6}&41.8&41.7&30.4&50.4&43.6&18.5&48&45.6&13.3\\\\
> MIRA&68.9&45.7&21.2&39.5&39.4&28.8&49&42.1&17.6&51.6&49.4&15.8\\\\
> CARP&70.3&48&23.9&\textbf{49}&\textbf{48.9}&\textbf{38.7}&\textbf{54.5}&\textbf{48.2}&\textbf{23.1}&\textbf{54.8}&\textbf{52.7}&\textbf{19.6}\\\\
> \hline
> \end{array}
>
> Overall, CARP outperforms MIRA and TWIST on clustering evaluation, mainly in transfer scenarios with more considerable differences in scores. Interestingly, TWIST outscores CARP on the ImageNet dataset. However, **its performance decreased significantly on transfer datasets/tasks such as Cifar-10/100 and GTSRB**. **CARP**, on the other hand, **scores strongly on all 4 datasets.** These results support the conclusion that CARP learns representations that generalize to future tasks.
>
> k-NN transfer learning (k=20), paper's Table 1
>
> \begin{array}{lrrrrrrrrrrrrr} \hline
> && Pets&Flowers&Aircraft&Cars&Country&Food&STL&GTSRB&Avg @k\\\\
> Method&Ep&&&&&&&&&10&20&100&200\\\\
> \hline
> TWIST&850&83.9&73.4&23.1&20.4&13.9&60.4&\textbf{96.6}&59.1&53.8&53.9&52.8&52.1\\\\
> MIRA&400&83.4&\textbf{81.4}&35.6&26.6&\textbf{14.3}&\textbf{64.2}&95.6&64.2&58.2&58.2&56.9&56\\\\
> CARP&400&\textbf{86.8}&80&\textbf{42.1}&\textbf{33.5}&12.3&58.4&95.9&\textbf{75.3}&\textbf{60.4}&\textbf{60.5}&\textbf{59.7}&\textbf{59.2}\\\\
> \hline
> \end{array}
>
> On average, *CARP k-NN transfer performance outweighs both MIRA and TWIST on the 8 transfer datasets.* Individually, CARP wins 4 out of 8 datasets.
>
> As pointed out by the reviewer, both MIRA and TWIST use the entropy term to avoid collapse during training. One important distinction, however, is that CARP maximizes the entropy over the random partitions and not over the entire set of prototypes. *We hypothesize (and show practical results in Table B2 in the appendix) that the random partition pretext task has a regularization effect that stabilizes training, avoids collapse, and improves final performance.* Because the subtasks constantly change, the model is less prone to collapsing all the embeddings into a single prototype. As a result, the regularization effect from the random partition diminishes the importance of entropy maximization. Hence, we do not need to consider tuning the entropy term's contribution to avoid trivial solutions. TWIST, on the other hand, needs to tune it carefully.
>
> > Given that all experiments are conducted using ResNet-50, experimental results with Vision Transformers (ViTs) shall strengthen the contributions of the paper.
>
> We thank the reviewer for the suggestion. It is in our plans to explore how the proposed random partition pretext task would behave with other architectures, such as ViTs. However, we chose ResNets as the main architecture mainly because most SSL methods provide baselines with ResNet backbones. Indeed, recent SSL methods are switching to ViTs. However, given our computing budget, running so many long experiments with more than one architecture would be impractical. For this reason, we decided to pick one base architecture and do the best we could regarding training, ablations, and evaluations to make our proposal clear and robust.
>
> > Why is it important or beneficial to avoid using non-differentiable modules, such as Sinkhorn-Knopp, for generating target cluster assignments? (L4-5, L99-100, L101-103)
>
> We apologize that the main text did not convey this matter clearly. We do not propose to avoid these types of non-differentiable modules, but rather we propose to explore a different solution. We will update our writing to convey this tone better.
>
> Our work provides an alternative solution to the cluster assignment problem with benefits and drawbacks. One benefit of having an end-to-end differentiable architecture might be performance-wise. The Sinkhorn-Knopp is an iterative algorithm that may require extra computation. Moreover, it also increases the number of hyperparameters that must be appropriately tuned. Nevertheless, solutions that employ the Sinkhorn-kopp deal with the extra compute and tuning and produce strong results. On the other hand, our approach shows that one can learn equally good, or even better, generalizable representations using a novel strategy that can avoid collapse in a deep learning integrated architecture.

---

> > ### Comment · Reviewer_VZFy · 2023-08-20
> >
> > I thank to authors for responding to my comments with additional experimental results. The authors' response and disccusions with other reviewers addressed most of my concerns about this work. I remain positive about this paper and maintain my score to accept.
> >
> > ---
> > I add more detailed comments to supplement my position.
> >
> > I believe that experiments with other architectures, such as ViTs, will strengthen the contribution and justification. The authors replied that they have plans to conduct experiments; i hope those plans will be made.
> >
> > For discussions and comparisons with related baselines, the authors give additional experimental comparisons in clustering and transfer learning evaluations. While the proposed method doesn't seem to outperform the baselines in all settings, it performs better on average. Although the MSN has not been experimented with, as stated by the authors, i agree that the proposed method has the advantage of not requiring additional hyperparameters (sharpening), loss, or careful tuning, unlike the baselines. This makes the proposed method attractive to me, even though the performance improvement is slight.
> >
> > I have read other discussions and comments on this page and agree that the idea is very simple, with a narrow and specific usage area, i.e., unsupervised clustering-based methods; thus, the contributions may seem limited. However, considering that there are many variants of works and techniques proposed to address the issue of collapsing, i believe the contribution is not insignificant.
> > The simplicity offered by the proposed method, which avoids introducing additional hyperparameters, sharpening, or speicialized loss functions, seems to me as an important contribution that distniguishes the method from others. While it is indeed possible to address stability and training through careful tuning, doing so may require multiple rounds of repetitive training and significant computational resources. This makes it difficult to apply these variant methods in various scenarios, e.g., computing budgets are limited.

---

> > > ### Author Response · Authors · 2023-08-21
> > > **Thanks to reviewer's work**
> > >
> > > We thank the reviewer for the fair assessment, valuable insights, and suggestions for our work. We will incorporate the relevant suggestions into the final version.

---

### Author Rebuttal · Authors · 2023-08-08

We thank the reviewers for their time reviewing our work and their valuable feedback. We will incorporate the additional results presented in this rebuttal as well as the suggestions in the final version of our manuscript.

In summary, our main contributions are the proposal of a stochastic partitioning process to improve the learned representations' robustness, as demonstrated in our comprehensive out-of-domain downstream evaluation. Moreover, we empirically show that our random partition pretext task produces a regularization effect that stabilizes the training and avoids collapse.

Our comprehensive evaluation protocol assesses the representation power of more than 11 SSL methods over 16 different datasets. We increased this evaluation, as requested, and included evaluation results for three additional methods, TWIST [a], MIRA [b], and CoKe [c]. The additional results are presented in the reviewers' individual replies.
In short, **CARP pre-trained representations remained top performant, mainly on transfer learning scenarios, which speaks for the power of the proposed random partition pretext task**.

In addition, we are providing an extra PDF document containing plots that strengthen the intuition about the sharpening effect of the consistency loss used by CARP.

We highlight that we evaluated our method against other ResNet-based models due to their prevalence in the existing SSL methods. In this light, our evaluation is fairer and demonstrates the improvements of the proposed partitioning while maintaining the backbones. While we share the desire to have more tests as proposed by the reviewers, due to our limited computing budget, we could not evaluate several backbones on our setup and ablations.

[a] Self-Supervised Learning by Estimating Twin Class Distributions, arxiv 2021

[b] Unsupervised Visual Representation Learning via Mutual Information Regularized Assignment, NeurIPS 2022

[c] Unsupervised Visual Representation Learning by Online Constrained K-Means. CVPR 2022.

---

### Decision · Program_Chairs · 2023-09-21

**Decision:**

Accept (poster)

**Comment:**

Summary
The paper presents an approach to improve clustering-based contrastive learning methods. The authors address the issue of collapse wherein all the data points map to the same point during training. The proposed idea (CARL) is simple and uses consistent assignments by way of random partitioning.

Strengths
- The proposed method addresses an important issue in self-supervised learning on collapse. The idea to achieve this by tweaking the partitioning is simple.
- The experimental setup in this work is solid. The results across image classification, retrieval, copy detection, few-shot evaluation show consistent improvements

Reviews & Justification
The paper received positive reviews overall. The only negative reviewer has concerns about relative contributions over CARL. After reading the rebuttal thoroughly, the AC believes that the authors have resolved this concern to the best extent possible. Since it is non-trivial to extend the original CARL method to ImageNet, the authors contributions over this work are significant.